

# Modelling floating riverine litter in the south-eastern Bay of Biscay: a regional distribution from a seasonal perspective

Irene Ruiz[1], Anna Rubio[1], Ana J. Abascal[2], Oihane C. Basurko[1]

[1]AZTI, Marine Research, Basque Research and Technology Alliance (BRTA), Pasaia, 20110, Spain

[2] IHCantabria - Instituto de Hidráulica Ambiental de la Universidad de Cantabria, Santander, 39011, Spain

*Correspondence to*: Irene Ruiz (iruiz@azti.es)

## Abstract

Although rivers contribute to the flux of litter to the coastal and marine environment, estimates of riverine litter amounts are scarce and the behaviour of riverine litter at river mouths and coastal waters is highly uncertain. This paper provides a comprehensive overview of the seasonal trends of floating riverine litter transport and fate in the south-eastern Bay of Biscay based on riverine litter characterization, drifters and high-frequency radars observations and Lagrangian simulations. Virtual particles were released close to the river mouths as a proxy of litter entering the ocean from rivers and were parameterized with a wind drag coefficient (Cd) to represent their trajectories and fate according to the buoyancy of the litter items. They were forced with numerical winds and measured currents provided by high-frequency radars covering selected seasonal week-long periods between 2009 and 2021. To gain a better insight on the type and buoyancy of the items, samples collected from a barrier placed at Deba river (Spain) were characterized at laboratory. Items were grouped into two categories: low buoyant items (objects not exposed to wind forcing e.g., plastic bags) and highly buoyant items (objects highly exposed to wind forcing, e.g., bottles). Overall, low buoyant items encompassed almost 90% by number and 68% by weight. Low buoyant items were parametrized with Cd=0%, and highly buoyant items with Cd=4%, this later one as a result of the joint analysis of modelled and observed trajectories of four satellite drifting buoys released at Adour (France), Deba (Spain) and Oria (Spain) river mouths. Results show that all regions in the study area are highly affected by rivers within or nearby the region itself. Simulations of riverine litter parametrized with Cd=4% showed that particles drifted faster towards the coast by the wind, notably during the first 24 hours. In summer, over the 97% of particles beached after one week of simulation. In autumn this value fell to 54%. In contrast, the low buoyant litter items take longer to arrive to the coastline, particularly during Spring with fewer than 25% of particles beached by the end of the simulations. When comparing coastline concentrations, the highest concentrations of particles (>200 particles/km) were recorded during summer in the French region of Pyrénées-Atlantiques for Cd=4%. These results coupled observations and a river-by-river modelling approach and can assist policy and decision makers on setting emergency responses to high fluxes of riverine litter arrivals and on defining future monitoring strategies for heavy polluted regions within the study area.





## 1 Introduction

Rivers act as key vectors bringing improperly disposed and mismanaged litter from land into coastal and marine environments, especially in densely populated or highly industrialized river basins. Riverine litter poses a large threat not only to coastal and marine environments but also to freshwater systems by degrading aquatic life, impacting freshwater quality and increasing economic losses associated with human activities (van Emmerik and Schwarz, 2020; Al-Zawaidah et al., 2021). Recent

findings derived from extensive modelling efforts suggest that about 1,600 rivers worldwide account for 80% of plastic inputs to the ocean with small urban rivers among the most polluting (Meijer et al., 2021). However, most of the litter research conducted to date has focused on marine environments (87%) when compared to freshwaters systems (13%), and only 7% of all scientific publications can be attributed to macroplastics (size > 2.5 cm) (Blettler et al., 2018). Riverine litter contributions to oceans are still uncertain, and results vary depending on the approach applied such as the dataset or the model used. Global

estimates based on modelled amounts of mismanaged plastic waste (MPW) range between 0.5 to 2.7 million metric tonnes per year (Lebreton et al., 2017; Schmidt et al., 2017; Meijer et al., 2021); however, they can represent less than a tenth when methodology followed differ from MPW-based models (Mai et al., 2020). Models require comprehensive field data and consistent and harmonized protocols to validate the amounts, type and size of riverine inputs, information that can then be used to implement tailor-made and effective measures at regional and local scale (González-Fernández and Hanke, 2017;

Wendt-Potthoff et al., 2020; Margenat et al., 2021). Such comprehensive data was obtained in Europe thanks to the RIMMEL project (González-Fernández and Hanke, 2017) and a network of visual observers of riverine macrolitter, which research concluded that between 307 and 925 million litter items are annually transferred into the ocean, mainly through small rivers, streams and coastal run-off (González-Fernández et al., 2021).

Once at the river mouth, riverine litter can accumulate nearby or it can move long distances, reaching remote areas from river

waters. Indeed, the distribution and fate of riverine litter in the coastal and marine environment is conditioned by the metocean conditions (currents, turbulence, wind) but also by the buoyancy of the objects, defined by their composition, size and shape (Ryan, 2015; Lebreton et al., 2019; Maclean et al., 2021). Objects with low buoyancy are mainly driven by currents contrary to high buoyant items which are pushed along the water surface partially by winds. The wind effect ("windage") is an important factor for pushing litter to shore and induce beaching, mainly for offshore-source litter, which is highly affected by winds,

compared to coastal-source macrolitter (Ko et al., 2020). Riverine litter trapped in near-shore areas is susceptible to beaching, settling and resurfacing episodes and reach open ocean mostly as small fragments (Morales-Caselles et al., 2021), hampering cleanup efforts and contributing to the prevalence of litter in the marine environment. Adjustment for windage has been consequently investigated in Lagrangian modelling studies in open ocean (Allshouse et al., 2017; Maximenko et al., 2018; Lebreton et al., 2019; Abascal et al., 2009) but also, although less mature, in coastal areas (Critchell and Lambrechts, 2016;

Utenhove, 2019; Tong et al., 2021). The lack of field data to accurately parametrize the effect of wind and validate simulation results is one of the key limitations both in riverine and marine transport modelling. From decades, researchers have used real observations derived from drifting buoys, such as in the Global Drifter program, which observations contribute to fill this gap.



Buoy data are used to fine-tuning prediction models and provide a better description of the near-surface circulation and its Lagrangian behaviour (Charria et al., 2013; Dagestad and Röhrs, 2019). They have also allowed simulating more realistic litter

pathways from origin to fate by integrating experimental windage parametrizations and the corresponding comparison between observed and modeled trajectories (Duhec et al., 2015; Pereiro et al., 2018; Rizal et al., 2021). Satellite-tracked drifting buoys and communication systems are costly, despite more economical and environmentally friendly solutions are gaining force among researchers. Examples include drifters built using biopolymers (Novelli et al., 2017; D'Asaro et al., 2020) and compact and lightweight designs with a GPS-tracking component for an easy deployment (Meyerjürgens et al., 2019b; van Sebille et

al., 2021). Others have evolved to develop drifters shaped as real litter items (e.g., plastic bottles), which allow a more accurate tracking position of standard objects, accounting for wind effect at sea and on inland waterways (Duncan et al., 2020).

Nowadays, coastal transport can be also characterized at high temporal and spatial resolution thanks to the use of land-based high frequency radar systems for the remote measurement of surface currents (hereafter HF radars (Rubio et al., 2017)). HF radars offer the opportunity to monitor surface currents in coastal areas, where the transport processes are significantly more

complex than open ocean waters due to the effect of coasts, bathymetry and other local forcings, like river discharges or coastal upwellings. Given the highly dynamic and complexity nature of coastal waters, this realistic and useful knowledge on coastal circulation combined with the parametrization of key physical processes affecting litter transport (e.g., windage) become crucial to reduce the uncertainties of modelled trajectories of riverine and marine litter (Van Sebille et al., 2020).

In the the south-eastern Bay of Biscay (hereafter SE Bay of Biscay), a HF radar provides, as part of the operational

oceanography system EuskOOS (https://www.euskoos.eus/), near-real-time surface current fields at 5 km spatial and 1-hour temporal resolution, covering since 2009 a range up to 150 km from the coast. This system has already been used in previously to study surface coastal transport processes in combination with multisource data (Manso-Narvarte et al., 2018, 2021; Rubio et al., 2011, 2013, 2018, 2020; Solabarrieta et al., 2014, 2015, 2016). The HF radar is also a good example of effective monitoring of surface currents with strong potential for floating marine litter management. The EuskOOS HF radar is part of

JERICO-RI (https://www.jerico-ri.eu/) and it is operated following JERICO-S3 project best practices, standards, and recommendations. Research conducted by Declerck et al., (2019) in the SE Bay of Biscay provided the first assessment of floating litter transport and distribution in the region, coupling surface currents observations from EuskOOS system, Lagrangian modelling and riverine inputs. Nowadays, these observations are used by local authorities both in real time and in hindcast in the framework of the operational service FML-TRACK (https://fmltrack.rivagesprotech.fr/) to collect floating

marine litter in the area. However, the accurate modelling of transport and fate of both floating marine and riverine litter need to consider the variety of floating objects and sources and additional physical processes as windage. This paper aims at estimating the seasonal trends on floating riverine litter transport and fate in the SE Bay of Biscay by modelling the Lagrangian behaviour of numerical particles released in the main rivers within the area. To do so, a Lagrangian model was forced by real observations from the EuskOOS HF radar and particles were parameterized to represent riverine litter trajectories according

to their observed buoyancy. Riverine litter collected from a local barrier was characterized at laboratory to explore the fraction of highly and low buoyant items. Since most of the items were low buoyant, simulations of particles considering only surface





currents were performed as the reference. Complementary Lagrangian simulations for highly buoyant items (and less abundant in the area) were also performed. In this case, 4 low-cost buoys with similar buoyancy of certain highly buoyant objects were built and released at 3 different rivers. Drifter data were used to parameterize the wind effect on this type of items and

consequently achieve more accurate results.

## 2 Study area

The study was conducted in the SE Bay of Biscay, between NE Spain (Basque Country) and SW France (Landes). The study area extends from 43.27°N to 44.58°N and from 3.18°W to 1.27°W, falling within the coverage area of the HF radar station of the operational oceanography system EuskOOS (Fig 1). The study area comprises two Basque regions - Bizkaia (Spain)

and Gipuzkoa (Spain) -, two French departments - Pyrénées-Atlantiques (France) and Landes (France) -, and eight rivers - Deba (Spain), Urola (Spain), Oria (Spain), Urumea (Spain), Oiartzun (Spain), Bidasoa (Spain), Nivelle (France) and Adour (France) -. The mean annual river discharge varies widely between rivers - 3.71 m$^3$/s (Oiartzun) to 350 m$^3$/s (Adour) (Sheppard, 2018). The bathymetry in the SE Bay of Biscay is characterized by the presence of a narrow continental shelf ranging between 7 and 24 km wide in the Basque area, gradually increasing along the French coast up to about 70 km (Bourillet et al., 2006;

Rodríguez et al., 2021). The continental shelf in the SE Bay of Biscay comprises two mainly areas, the Aquitaine shelf with a N-S orientation and Cantabrian shelf with an E-W orientation. The continental slope is very pronounced, with a slope of the order up to 10%‐12% (Sheppard, 2018).

The circulation of the self-water masses is marked by a seasonal variability. At shorter temporal scales, circulation is mostly modulated by the bathymetry and the coastal orientation, the density-driven currents, and winds (Le Boyer et al., 2013;

Solabarrieta et al., 2014). Tidal currents in the area are quite week constrained by topography and width on the continental shelf (Lavin et al., 2006; González et al., 2007; Karagiorgos et al., 2020). Along-shelf currents are more intense and persistent during winter and autumn (about 10–15 cm s$^{-1}$), contrary to the other seasons, especially in summer (about 2.5 cm s$^{-1}$)(Charria et al., 2013). In winter, the prevailing SW winds causes an E to N flow and the moderate to strong NW winds occurring in spring and summer induce S and SW surface currents circulation over the French and Spanish coasts accompanied by a greater

variability (Solabarrieta et al., 2015). In winter, westerly winds in the Basque coast reinforce the slope current (named "*Iberian Poleward Current*" (IPC)), a warm and saline intrusion trapped within the 50 km of the shelf edge, achieving its greatest velocities (up to 70 cm s-1) during this season. The IPC favours the along slope transport of water masses (Solabarrieta et al., 2014; Porter et al., 2016). The exchange between shelf and deep sea waters in winter is associated to the generation of eddies, from the interaction of currents with the topography (Lavin et al., 2006; Rubio et al., 2018; Teles-Machado et al., 2016).

Maximum run-offs combined with SW winds also allow river plumes spread northwards and along the French shore during winter. However, this path changes in spring, when river discharges are reduced and winds blow from NW (Lavin et al., 2006; Puillat et al., 2006). The main circulation features in the study area are summarized in the figure created by (Declerck et al., 2019).

Floating marine litter distribution in the SE Bay of Biscay follows the general circulation in the area, showing a high retention

during spring and summer and a northward dispersion along the French coast during autumn and winter (Declerck et al., 2019;



Rubio et al., 2020). Longer residence times and higher concentrations are observed in winter influenced by the run-offs and the influx of floating litter from local but also from distant sources (Pereiro et al., 2019). In the Bay of Biscay, macrolitter with high windage tends to accumulate in nearshore areas (with a probability of around 90%) or are beached (with a probability higher than 60%) (Rodríguez-Díaz et al., 2020). A similar trend was observed in the study area by (Ruiz et al., 2022) who concluded that macrolitter items with a high windage rapidly beach during spring and summer underlining the importance of windage effect on the coastal accumulation. Since June 2020, innovative detection and tracking solutions combining ocean modelling and remote observation systems are operating in the SE Bay of Biscay for support floating marine litter reduction strategies both downstream (interception at sea with collect vessels and on beaches with cleaning facilities) and upstream (source identification and reduction) (Delpey et al., 2021).

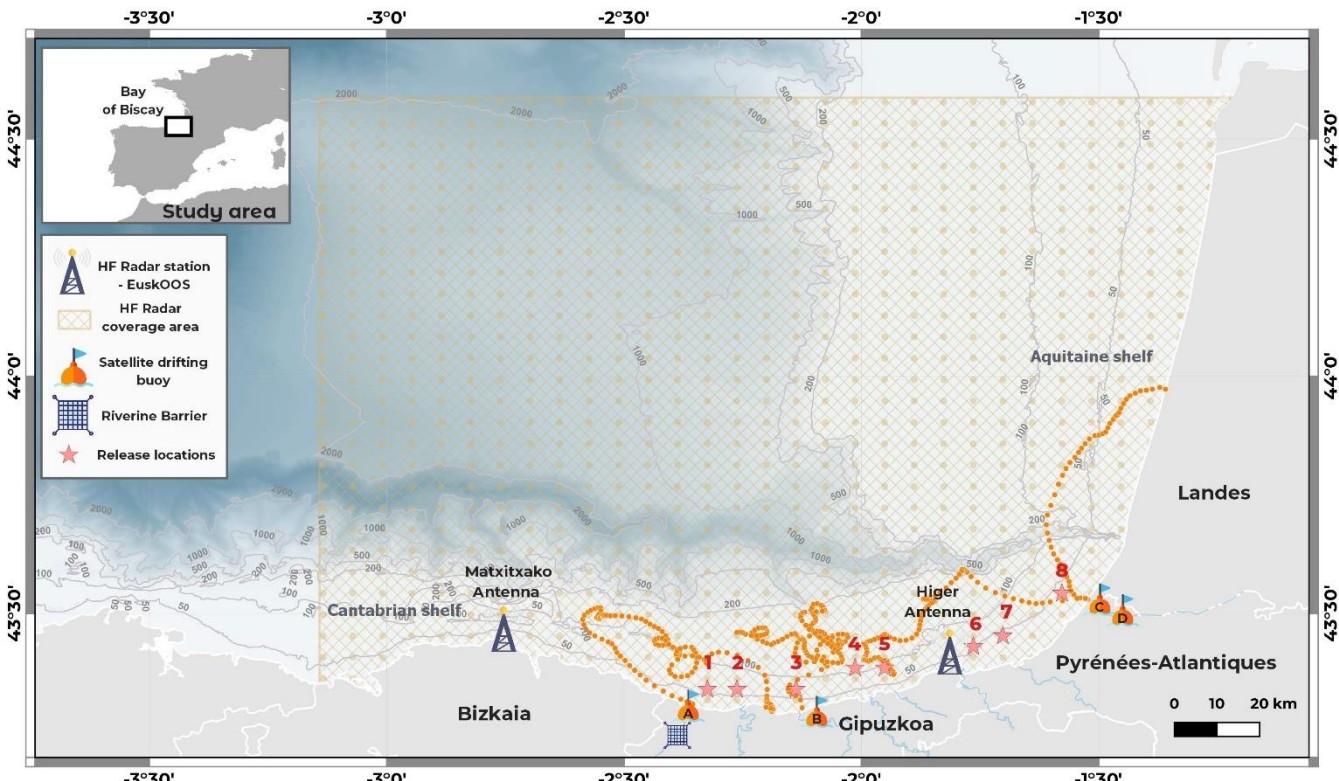

Fig 1. Study area with the release locations of the Satellite drifting buoys and the riverine barrier. Dots in orange represent the trajectories of the buoys. Numbers correspond to the particle releasing location for riverine litter simulations: (1) Deba; (2) Urola; (3) Oria; (4) Urumea; (5) Oiartzun; (6) Bidasoa; (7) Nivelle; and (8) Adour River. Dots in light yellow represent the nodes of the HF Radar grid. (Basemap - EMODnet Bathymetry portal: www.emodnet-bathymetry.eu)

## 3 Methods and Data

### 3.1 Riverine Litter Sampling

In Spring 2018, a riverine barrier was placed in Deba river (Gipuzkoa) to retain and collect floating macro riverine litter during low to moderate flows. The barrier, which consisted of a nylon artisanal net supported by hard floats (buoys) was 40 m long and 0.6 m high with a 60 mm mesh size (see photos in Appendix A). A sampling was conducted weekly from April 2018 to





June 2018; in total eight riverine litter samples were collected. Litter items were quantified, weighted, and categorized at lab according to the Master list included in the "Guidance on Monitoring of Marine Litter in European Seas" (Galgani et al., 2013) Items were grouped into 7 types of material (artificial polymer materials, rubber, cloth/textile, processed/worked wood, paper/cardboard, metal, and glass/ceramics) and further classified into 44 categories (see the classification in Appendix B). Riverine litter items were also categorized into two groups (low and highly buoyant items) considering their exposure to wind

based on (Ruiz et al., 2022).

### 3.2 Drifters Observations

Four satellite drifting buoys (herein after 'low-cost buoys') were built by the authors and deployed one-by-one in the river mouths of Oria (1 buoy), Deba (1 buoy), and Adour (2 buoys) between April 2018 and November 2018 (Fig 1, Table 1). The 'low-cost buoys' provided positioning every 5 minutes using satellite technology. 'Low-cost buoys' were 9 cm in height, 9.5

cm in float diameter and weighed approximately 200 g (Fig 2). A GPS (SPOT Trace device) was placed in the bottom of a high-density polyethylene HDPE plastic container sealed to guarantee water tightness. Almost 2/3 of the buoy floated above the water surface thus preventing any satellite signal losses. Transmission periods relied upon battery lifetime and buoys landing.

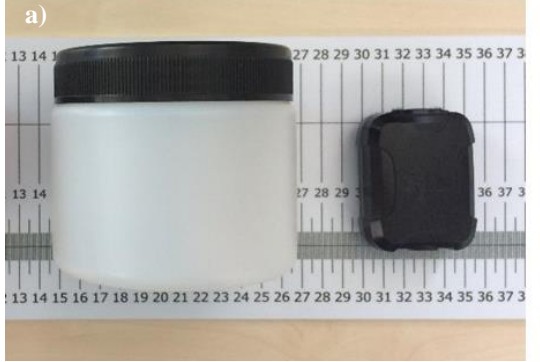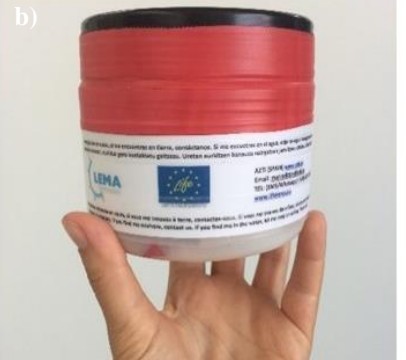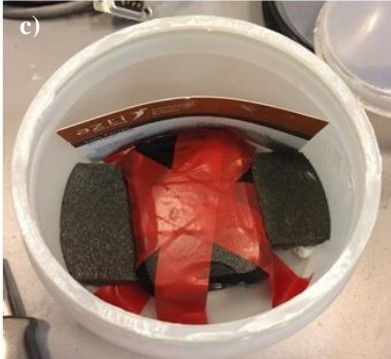


Fig 2. Main components of the "Low-cost buoy". The structure: (a) HDPE container and SPOT Trace device powered by 4 AAA cells. Assembly process: (b) final appearance once the buoy is sealed. The buoy is labelled with contact information both within and outside; (c) the SPOT Trace was fixed at the base of the container with adhesive tape to avoid twists and turns of the buoy.

Table 1. Locations, periods, and distances covered by the drifting buoys

| Buoy ID | River | Initial date | Final date | Distance covered (km) |
|---|---|---|---|---|
| A | Deba | 16-Sept-2018 8:00 | 4-Oct-018 7:00 | 116.1 |
| B | Oria | 12- Apr-2018 16:00 | 18-Apr-2018 12:00 | 118.72 |
| C | Adour | 29-Jul-2018 20:00 | 2-Aug-2018 20:00 | 71.21 |
| D | Adour | 28-Nov-2018 9:00 | 30-Nov-2018 11:00 | 64.41 |



### 3.3 HF radar Current Observations and wind data

Surface velocity current fields were obtained from the EuskOOS HF radar station composed by two antennas located at
Matxitxako and Higer Capes and covering the SE Bay of Biscay (see (Solabarrieta et al., 2016; Rubio et al., 2018) for details)..
Data consist of hourly current fields with a 5 km spatial resolution obtained from using the gap-filling OMA methodology
(Kaplan and Lekien, 2007; Solabarrieta et al., 2021). Data used for the Lagrangian simulations were extracted considering the
outputs from the standard QC (quality control) procedures for real-time HF radar data (Rubio et al., 2021). Once extracted,
data were visually inspected to ensure a complete radial coverage (i.e., ensuring optimal OMA reconstructed fields) and build
data subsets for the Lagrangian simulations avoiding periods with temporal gaps of more than a few hours.
Hourly ERA5-U10-wind fields were obtained from the atmospheric reanalysis computed using the IFS model of the European
Center for Medium-Range Weather Forecast (ECMWF) (see (C3S, 2019) for details). ERA5 atmospheric database covers the
Earth on a 30 km horizontal grid using 137 vertical levels from the surface up to a height of 80 km and provides estimates of
a large number of atmospheric, land and oceanic climate variables on a $0.3° \times 0.3°$ grid, currently from 1979 to within 3 months
of real time. Both HF radar current observations and wind data cover the drifter's emission periods and the selected week-long
periods between 2009 and 2021 for riverine litter simulations.

### 3.5 Particle Transport Model

The transport module of the TESEO particle-tracking model (Abascal et al., 2007, 2017a, b; Chiri et al., 2020) was applied to
simulate the transport and fate of riverine litter items from selected rivers once they arrive to the coastal area. Simulations were
forced by HF radar surface current velocity and wind data. The transport module was also used to accurately estimate the
windage coefficient by calibrating the model according to the 'low-cost buoys' trajectories. TESEO has been calibrated and
validated by comparing virtual particle trajectories to observed surface drifter trajectories at regional and local scale (Abascal
et al., 2009, 2017a, b; Chiri et al., 2019). Although the TESEO is a 3D numerical model conceived to simulate the transport
and degradation of hydrocarbons, it has also been successfully applied to other applications such as the study of transport and
accumulation of marine litter in estuaries (Mazarrasa et al., 2019; Núñez et al., 2019) and in open waters (Ruiz et al., 2022).

### 3.5.1 Wind drag estimation

Two simulation strategies were combined for (1) estimating the wind drag coefficient and (2) study the seasonal behaviour of
riverine litter items in the area (section 3.5.2) The wind drag coefficient (Cd) was determined by comparing the observed
trajectories provided by the 'low-cost buoys' and the modelled trajectories performed with TESEO. The test was done through
different parametrizations of the wind drag coefficient  ranging from 0% to 7% (Table 2). This range was chosen based on
previously  floating marine litter studies coupling Lagrangian modelling and observations from satellite drifting buoys (Carson
et al., 2013; Stanev et al., 2019; Van Der Mheen et al., 2019). The coefficient providing the lowest error was considered the
best coefficient to simulate highly buoyant litter. Due to the grid limitations of the surface currents and wind data in the coastal



area, the comparison was not initialised at the launching position of the 'low-cost buoys' (river mouths) but instead it was

initialised at the closest grid element that contained valid currents and wind data (Table 1). Observed positions were

interpolated onto a uniform one-hour time, fitting the met-ocean temporal resolution. A release of 1,000 virtual particles was

performed every 4 hours at the corresponding observed position (Table 2). Particles were tracked over a 24-hour period and

the trajectory of the center of mass of all the particles was computed at every time step to represent the track of the particle

cloud. Observations were compared to modeled trajectories using the simple separation distance, which is the difference

between the observed and the computed position of the center of mass at a time step t. Mean separation distance $\overline{D(t^{mod})}$ was

calculated for every modelled position based on the simple separation distance following Eq. (1):

$$\overline{D(t^{mod})} = \frac{1}{N}\sum_{i=1}^{N}\left|\vec{X}^{mod}(t^{mod}) - \vec{X}^{obs}(t^{obs})\right| \quad (1)$$

where $\vec{X}^{mod}(t^{mod})$ and $\vec{X}^{obs}(t^{obs})$ are the modeled and observed trajectories for the simulation period i of a total of N

periods. A mean separation distance curve was computed for every wind drag coefficient derived from the mean separation

distance curves of the four buoys. The area beneath the mean separation distance curve was calculated to select the more

suitable wind drag coefficient. The area $\widetilde{D}$ was calculated as a numerical integration over the forecast period via the trapezoidal

method following Eq. (2):

$$\widetilde{D} \approx \int_{t^{mod}=1}^{t^{mod}=24}\overline{D(t^{mod})}dt \quad (2)$$

**3.5.2 Lagrangian seasonal simulation of riverine litter items**

Seasonal simulations were run for low and highly buoyant items to assess the seasonal differences on riverine litter transport

and fate. As parametrizations concerning wind effect linked to the object characteristics are scarce, the optimal wind drag

coefficient estimated for the buoys (see section 3.5.1) was accounted for simulated the behaviour of the objects highly exposed

to wind. No wind drag parametrization (Cd=0%) was applied for low buoyant objects not subjected to wind effect. A total of

ten periods per season uniformly distributed within the study period (2009-2021) were considered for the simulations based

on the availability of HF radar surface current datasets (see Appendix C for the selected periods). In total, 4,000 particles were

released in 8 rivers for each selected period (500 per river) (Table 2). Simulations were run for 7 days. The total number of

particles modeled for Cd=0% was the same as Cd=4%. Particles were released around 2.5 nautical miles off the coastline due

to the complexity in resolving small-scale processes in and near the river mouths. A post-processing was carried out to compute

by river: (1) the particles evolution over the time from their release until their arrival to the coastline; and (2) the particles

distribution on the coastline, counting the number of beached particles per km of coastline and indicating the spatial

concentration per region.



Table 2. Simulation, release, and physical parameter values for wind drag estimation and floating riverine litter simulations.

| | Simulation parameters | | | Release parameters | | Physical parameters | |
|---|---|---|---|---|---|---|---|
| | Number of particles | Integration time | Time step | Release locations | Release time | Turbulent diffusion coefficient | Wind drag coefficient (Cd) |
| **Simulations for wind drag estimation** | 1,000 per location | 24 h | 60 s | At the observed locations of the buoy | Over the emitting period of the buoy at spaced intervals of 4 hours | 1 m²/s | 0 %, 2%, 3%, 4%, 5%, 6% , 7% |
| **Seasonal riverine litter simulations** | 500 per river | 1 week | 60 s | At a distance of 2.5 nautical miles from the river mouth | At the beginning of the selected time period (10 periods per season) | 1 m²/s | 0 %, 4% |


## 4 Results

### 4.1 Riverine litter characterization

In total 1,576 items and 11.597 kg of riverine litter were sampled and characterised (Fig 3). *Plastic* was the most common type of riverine litter in terms of number of items (95.1%) and in weight (67.9%); they were also frequent *Glass/ceramics* (16.1%)

and *Cloth/textile* items (6.9%) when counted by weight. The top ten litter items accounted for 93.3% by number and 72.6% by weight of the total riverine litter (Table 3). *Plastic/polystyrene pieces between 2.5 cm and 50 cm* and *Other Plastic/polystyrene identifiable items* (e.g., food labelling) were the most abundant in terms of number (71.2%) and weight (16.9%). *Low buoyant items* encompassed almost 91% by number and 68% by weight of litter items (Fig 4).

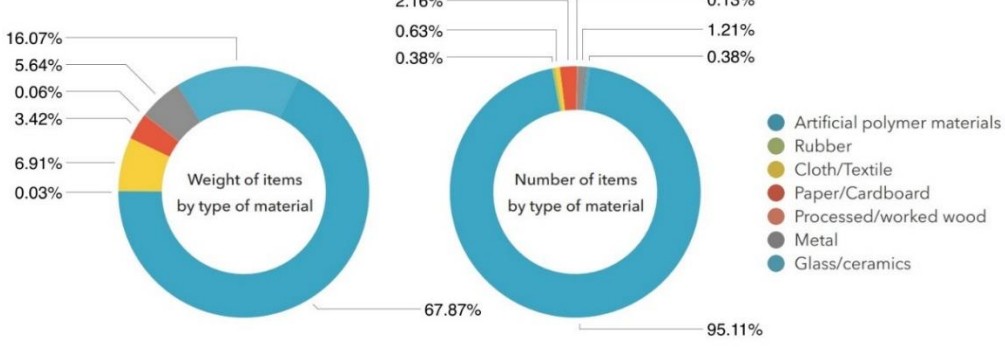


Fig 3. Composition by type of material based on the number and weight of riverine litter items collected in the riverine barrier located in Deba river (Gipuzkoa) between April and June 2018.



Table 3. Top ten (X) riverine litter items collected in the riverine barrier located in Deba river (Gipuzkoa) between April and June 2018. Items have been ranked by abundance (left) and weight (right) according to the MSFD Master List Categories of Beach Litter Item and classified based on their exposure to wind effect.

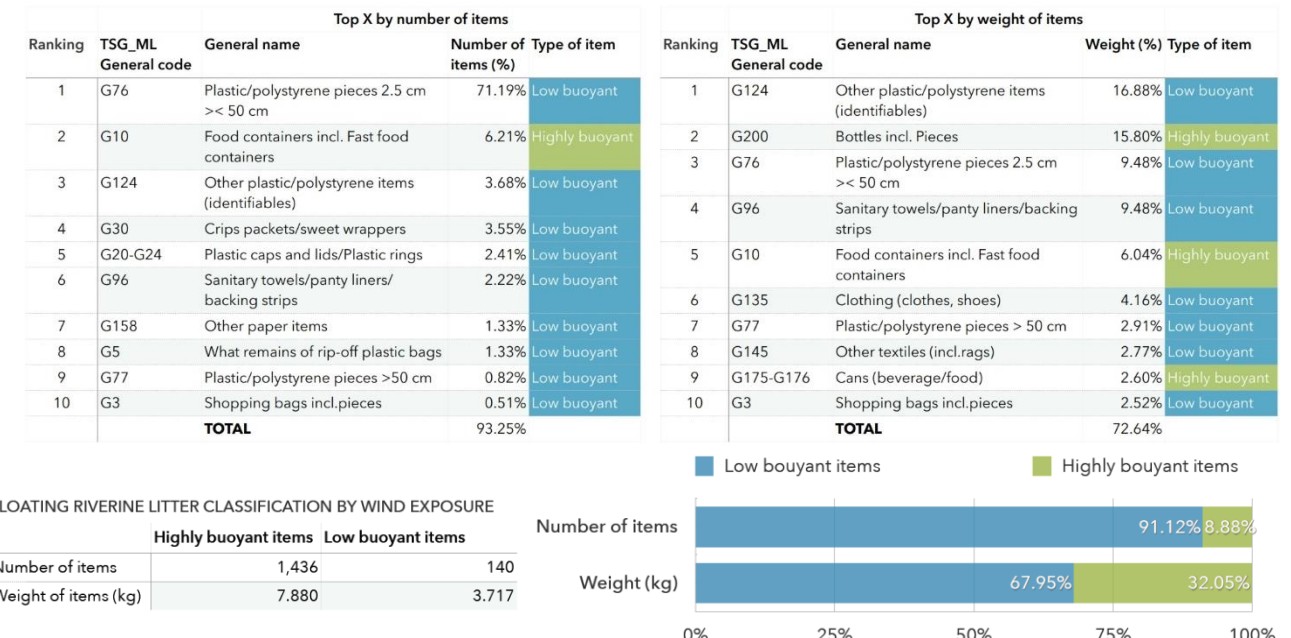

Fig 4. Riverine litter items classification based on the exposure to wind effect, from riverine litter items collected in the riverine barrier located in Deba river (Gipuzkoa) between April and June 2018.

## 4.2 Wind drag coefficient for drifting buoys

Total distances covered by drifting buoys ranged from 62 km to 118 km (Table 1) and they all spread out over the rivers inside the HF radar coverage area, spanning approximately 44ºN and 2º 22'W. They provided position data over 385 h before beached on Landes and Gipuzkoa coastlines. When compared with numerical trajectories obtained using different Cd parameterizations, the mean separation distance ($\overline{D(t^{mod})}$) increased nearly linearly with time for all the parametrizations, achieving a maximum separation of almost 14 km at 24 hours for Cd=0% (Fig 5). Overall, using no windage parametrization gave the largest $\overline{\overline{D}}$. Simulations parametrized with Cd=4% gave the best results with an average ± standard deviation (SD) of 3.2 ± 1.25 km and a maximum value of 4.85 km at 24 h. When assessing the mean separation distance for all the modeled positions at every observed position of the buoys, the most common range separation distance for Cd=4% was 2- 4 km (Fig 6). Hence, a wind drag coefficient of 4% was applied in the remaining analysis to estimate riverine litter behaviour of highly buoyant items.

## 4.3 Seasonal trends on floating riverine litter transport and fate

Particle **concentrations** in the coastline varied between 0 and 258.46 particles/km (Fig 7). Particles parametrized with Cd=4% drifted faster towards the coast by the wind, notably during the first 24 hours. The highest concentrations (>200 particles/km) were recorded during summer in Pyrénées-Atlantiques for Cd=4%, probably due to the seasonal retention patterns within the




study area (Appendix D). Although less intensely, Cd=4% also lead to a high particle concentration in Pyrénées-Atlantiques (106.86 particles/km) and Gipuzkoa (166.1 particles/km) during winter. Lowest concentrations (0-20 particles/km) were recorded for Cd=0% at all seasons during the first 24 hours and particularly during autumn. Overall, Bizkaia was the less
affected by litter for both windage coefficients (<40 particles/km). When looking at the total amounts of **beached particles per season**, in summer over the 97% of particles parametrized with Cd=4% beached after one week of simulation (Fig 8). In autumn this value fell to 54%. In contrast, particles parametrized with Cd=0% take longer to arrive to the coastline, particularly during Spring with fewer than 25% of particles beached by the end of the simulations. According to the **temporal evolution** of floating particles released per river, particles beached remarkably fast within the first 24-48 hours for Cd=4%, particularly
those released during summer in French rivers. Similar behaviour pattern was observed within the same season between rivers, probably influenced by the vicinity of rivers and the spatiotemporal resolution of forcings (Fig 9). When looking the **seasonal trends** by river and region, beached particles were mainly found in Gipuzkoa for both Cd=4% and Cd=0% - 40.1% and 11.54% of the total particles released respectively -, particularly in winter after one-week of simulations. For Cd=0%, beaching from particles released in Bidasoa, Nivelle and Adour River was higher in summer (9.01% particles released during summer) though
this trend was reversed in autumn, when particles released in Basque rivers resulted in higher beaching. Overall, all regions were highly affected by rivers within or nearby the region itself (Fig 10).





Figure 5. Mean separation distance between modelled and observed trajectories for each wind drag coefficient. The dark line is the mean curve employed
for the trapezoidal integration.







Figure 6. Spatial mean distance between modeled and observed trajectories of buoy A, B, C and D with a drag coefficient Cd=4%. Particle trajectories were simulated during 24 h, with a re-initialization period every 4 hours. The modeled trajectories are shown in solid lines. Circles represents at the observed position the mean separation distance for all the modeled position



Figure 7. Particle concentration in Bizkaia, Gipuzkoa, Pyrénées-Atlantiques and Landes coastlines. The seasonal distribution is shown for wind drag coefficient Cd=0% and Cd=4% after 24 hours and 168 hours of simulation




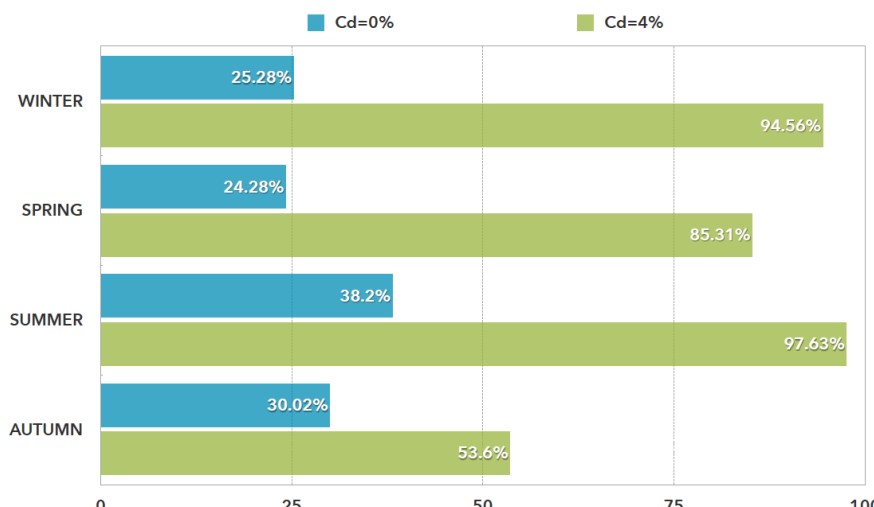

Figure 8. Total amounts of beached particles per season after 168 hours of simulation for wind drag coefficient Cd=0% and Cd=4%.

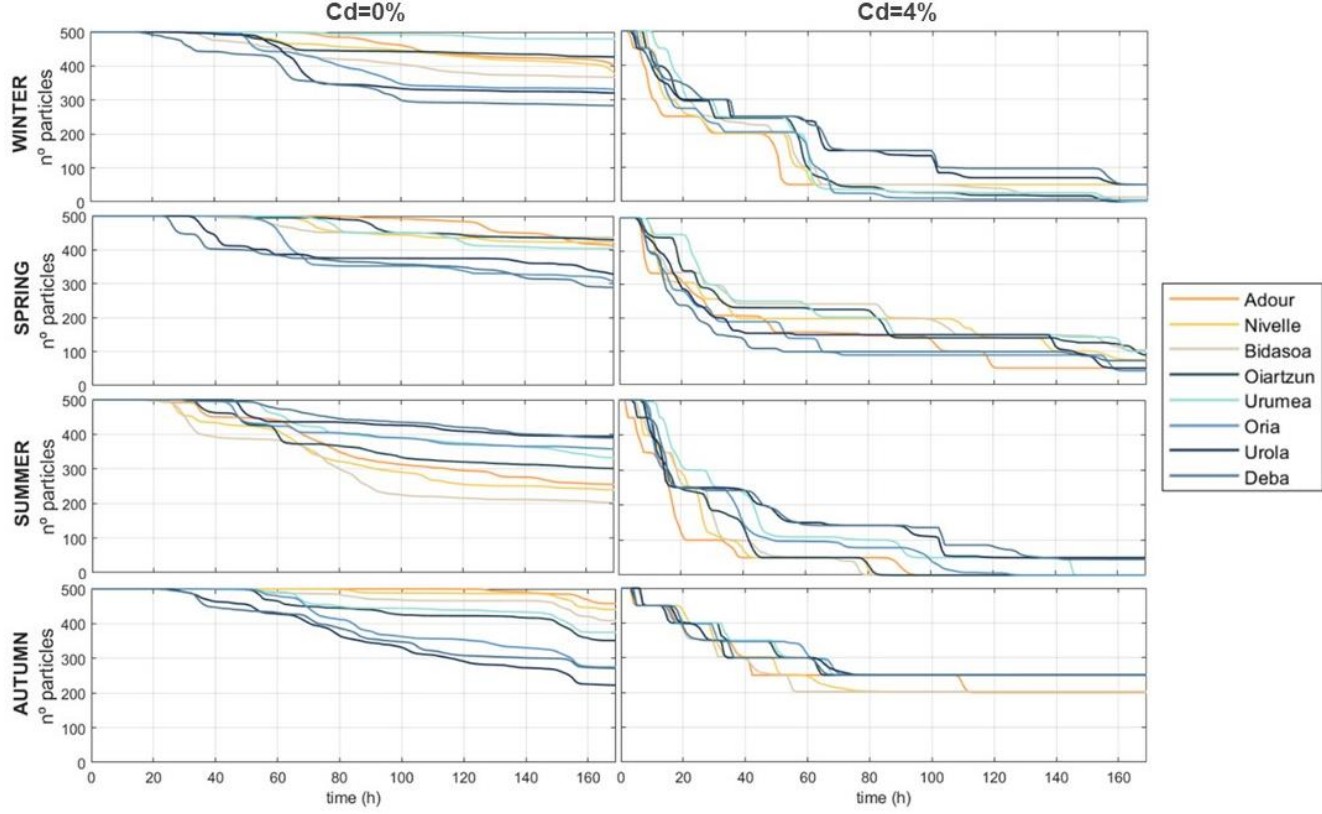

Figure 9. Temporal evolution of the particles released by river during the simulation period for a wind drag coefficient Cd=0% and Cd=4%. The curves represent the number of floating particles in the water surface for every time step.





Figure 10. Seasonal analysis of beached particles per region and river for wind drag coefficient Cd=0% and Cd=4% by the end of the simulation period. The nodes of the region correspond to the number of beached particles. The width of the node depicts the sum of the beached particles, and the links represent the number of particles beached per river.




## 5 Discussion

### 5.1 Riverine litter composition

In this study, an artisanal net placed at the mouth of Deba river provided a practical and tailored application for aggregating
riverine in the study area during Spring 2018. Short and narrow rivers prevail in the SE Bay of Biscay particularly affected by
a strong tidal regime, and very intense, stationary and persistent storms caused by a combination of a warm sea, an unstable
surface atmosphere and cold air at higher altitudes (Ocio et al., 2015). First field studies aiming at reporting the abundance and
composition of floating riverine litter in European rivers date back less than 10 years and they were performed mainly in larger
and more abundant rivers than Deba river. Despite the morphology and hydrological differences between rivers, the
distribution of items by type of material in Deba river showed a clear predominance of plastic as observed in Siene (Gasperi
et al., 2014), Danube (Lechner et al., 2014) or Rhine River (van der Wal et al., 2015). Similarities were also found when
comparing the Top ten list of riverine litter items to rivers located in the North-East Atlantic region. *Plastic/polystyrene pieces
between 2.5 cm and 50 cm* top the list in terms of number of items, accounting for a greater proportion in Deba river (71.2%)
than in North-East Atlantic rivers (54.53%)(Bruge et al., 2018; Gonzalez-Fernandez et al., 2018). Riverine litter items trapped
on vegetation or deposited on the riverbank can be degraded by weather conditions (rain, wind, etc.) favouring the
fragmentation in plastic pieces before their arrival to the coastal and marine environment. Higher percentages of
*Plastic/polystyrene pieces between 2.5 cm and 50 cm* observed in the study than those of the Black Sea (13.74%) or the
Mediterranean Sea (25.01%) can be attributed to a higher and faster fragmentation of riverine items along Deba river and the
North-East Atlantic basins. Results are also in line with the ranking list of the Top ten beach litter items across the North-East
Atlantic region revealing that Single Use Plastics (i.e. food containers, bottles and other packaging) are among the most
abundant riverine litter items together with plastic fragments (Addamo et al., 2017). These results differed from the analysis
performed in sea small-scale convergence areas of floating marine litter ("*litter windrows*") on the coastal waters of the SE
Bay of Biscay, where fishing-related items were the second most abundant sub-category in terms of number after
*Plastic/polystyrene pieces between 2.5 cm and 50 cm* (Ruiz et al., 2020a). Substantial differences also exist between riverine
litter sampled in Deba river and floating marine litter assessed by visual observation from research vessels in open waters of
the Bay of Biscay (Ruiz et al., 2022). Differences might be related to the monitoring method and, also, to the size of the items,
since small items, as plastic pieces, can be overlooked by the observer when visual counting method is applied, contrary to
riverine litter samplings for later analysis at lab. Overall, riverine litter data acquisition is mainly focused on the floating
fraction and the litter loads under the surface water are often ignore. Increasing the quantity of rivers sampled, the frequency
and the riverine water compartments is necessary to establish the composition and trends of riverine litter in the SE Bay of
Biscay.

### 5.2 Wind drag estimation

One of the largest uncertainties for simulating floating litter behaviour is the proper quantification of a wind drag coefficient.
Empirical data provided by "Low-cost buoys" combined with surface current measurements by HF radar can be used as a
proxy for predict the drift of floating litter objects with similar buoy characteristics (density, size and shape) in the study area.
Commercial SPOT Trace devices have been used over the past few years in coastal and open ocean applications in a wide
range of studies. Studies range from calibrating HF radars (Martínez Fernández et al., 2021), tracking drifting objects as
icebergs (Carlson et al., 2020), pelagic Sargassum (Putman et al., 2020; van Sebille et al., 2021) or fishing vessels
(Widyatmoko et al., 2021; Hoenner et al., 2022), to search and rescue training (Russell, 2017) and oil spill and litter monitoring
(Novelli et al., 2018; Meyerjürgens et al., 2019a; Mínguez et al., 2012; Abascal et al., 2015). Nevertheless, object
characteristics may change over the time due to the exposure to wind, waves, UV radiation, seawater and the attachment of





organic material (Kooi et al., 2017; Min et al., 2020). Objects become breakable, and biofouling increases their density, overcoming the positive buoyancy and impacting on their trajectory. Investigations so far pinpointed longer time scales (weeks to months, and lager) than considered in this study (days) for a significant change on the behaviour of floating objects (Ryan,

2015; Fazey and Ryan, 2016). Consequently, physical variations on the buoy properties were not accounted for wind drag estimation. The separation distance between observed and modeled trajectories has been commonly used to evaluate the skill of particle-tracking models (Callies et al., 2017; Haza et al., 2019; Aksamit et al., 2020; Abascal et al., 2012). In this study, the purpose was no to evaluate the model accuracy but estimated the wind drag coefficient for the "Low-cost buoys". However, the novel approach proposed by (Révelard et al., 2021) may be of particular interest for future experiments oriented to assess

the wind drag coefficient of highly buoyant items drifting during short time periods in the coastal area. The results obtained for Cd=4% can be consistent with wind drag estimations for the Bay of Biscay of the partially emerged *Physalia physalis* (Ferrer and Pastor, 2017) but greater than the Cd=3% observed for the Prestige oil spill accident (Abascal et al., 2009; Marta-Almeida et al., 2013). Indeed, oil spill studies refer to a range of wind drag coefficient between 2.5 to 4.4% of the wind speed, with a mean value of 3 - 3.5% (e.g., ASCE, 1996; Reed et al., 1994). In this study, a wind drag value of 4% can be expected

due to the strong buoyancy of the "low-cost buoys" and can be applied for simulating the transport and fate of a specific group of litter items that share similar characteristics. However, due to the large heterogeneity of highly buoyant items, further experiments are needed to better parametrize the wind drag coefficient of different objects and consequently reduce the uncertainties on their behaviour.

### 5.3 Seasonal riverine litter distribution by region

It is broadly accepted that the SE Bay of Biscay is polluted with floating litter discarded or lost at the marine and coastal area but also with litter originated inland and transported via rivers and runoff. However, detailed studies on riverine litter contribution are still scarce and modelling efforts combining observations and physical parametrizations of riverine litter properties are non-existent. This study shows that the exposure to wind effect of riverine objects largely control their transport and coastal accumulation in the SE Bay of Biscay, with concentrations varying between regions and over the time.

Concentrations in Pyrénées-Atlantiques and Gipuzkoa regions diverged widely from the other studied regions. Indeed, the highest concentrations occurred in both regions during summer for low buoyant (100-120 particle/km) and but also for highly buoyant items (>200 particles/km). Although larger amounts of particles beached in Gipuzkoa during summer, concentrations are lower than Pyrénées-Atlantiques since the coastline in the Basque region is longer. Low buoyant pathways and fate reflect the well-known surface water circulation patterns in the SE Bay of Biscay. Concentrations of floating riverine litter are

therefore a direct consequence of the seasonal variability of floating drift and results are in line with findings provided by (Declerck et al., 2019) who pinpointed a higher coastal retention in the area during spring and summer. Low buoyant objects not subjected to windage effects remain floating at the coastal waters and highly buoyant objects tended to beach remarkably faster as reported in literature by (Rodríguez-Díaz et al., 2020). However, long-term data collected by in-situ observations of beached litter across the different regions are necessary to validate the large seasonal variations and to assess the reliability of

concentration levels for addressing riverine litter issue in priority regions with heavily polluted coastlines.

### 5.4 Rivers as key vectors of riverine litter

The interpretation of the spatial and temporal riverine litter distribution by river can be challenging since riverine litter fluxes in the study area are highly uncertain. In the study area, two major assumptions were made regarding the river systems: (1) same river discharge for all rivers and (2) same river discharge for all seasons. This means that same amounts of riverine litter

were allocated for every river regardless the differences on the width and depth and the seasonal flow variations. Since each river basin has its own particularities, future modelling approaches should be adapted to the the morphology and hydrological





conditions of the catchment area. Other drivers as the land use or socio-economic factors such economic status or population density can be a determining factor on the amount of mismanaged litter that could contribute to riverine litter fluxes (Schmidt et al., 2017; Schuyler et al., 2021). It is also necessary to further investigate if higher river flows in the area are directly related

to an increased discharge of riverine litter since analysis already performed in different river basins show contradicting relations between the occurrence of riverine litter and river fluxes (van Emmerik and Schwarz, 2020). Along with the complex nature of qualifying riverine litter fluxes, litter behaviour in the coastal area of the SE Bay of Biscay is still in its early stage, and much has yet to be revealed. Particular attention should be paid to Pyrénées-Atlantiques and Gipuzkoa, as main impacted regions in the studied area. The dominant number of rivers in this region can favour accumulation trends regardless the season.

Regional coordination should be reinforced due to the transboundary movement of riverine litter in the study area and reasonable efforts oriented to retain or remove riverine litter as clean-up measures in the riverbanks should be investigated to avoid litter being transported to the coastal and marine environment.

### 5.5 Model limitations

The coastline of the SE Bay of Biscay is mainly covered by sand and muddy-sand and characterized by the presence of

moderate to high sea rocky cliffs, especially in the Basque region (ICES, 2019; Bilbao-Lasa et al., 2020). The geomorphology can affect the retention of litter washing ashore. Sandy beaches tend to be more efficient at trapping and thus accumulating litter than rocky areas which favor litter fragmentation (Robbe et al., 2021; Weideman et al., 2020). Waves and tides can also constrain coastal accumulation since they can resuspend litter and transport it back into the ocean (Brennan et al., 2018; Compa et al., 2022). Nevertheless, research on these processes is scarce and they cannot be resolved yet at a suitable resolution (Melvin

et al., 2021). Consequently, in this study once particles beached, they were classified as it arrived to their final destination. It is, however, important to consider for future research in the study area the link between coastal accumulation, and the type of shoreline and resuspension, even though the model cannot yet simulate these processes. The release location strongly influences where litter accumulates on the coastline. Litter items can beach rapidly when release locations are located near the coastline (Critchell et al., 2015). However, there is a big gap between the spatial resolution of ocean circulation models (up to

10 km spatial resolution) and the complex coastal accumulation processes. In this study, the release locations were located distant for the sources to avoid uncertainties on model performance at smaller scales. However, a greater model resolution with a finer grid can reinforce simulation results (NOAA, 2016). Nested models, flowing from fine resolution near critical locations as the river mouths to open ocean resolution is a worthy issue for future consideration.

### 5.6 Riverine litter collection and monitoring by a floating barrier

Riverine litter quantities on a global scale urge countries to keep rivers pollution-free, intercepting riverine litter before it reaches the ocean and minimizing the impact of marine pollution from land-based sources. Research to date suggest that a significant reduction of marine litter in the ocean can be achieved with collection at rivers or with a combination of river barriers and clean up ocean devices (Hohn et al., 2020). Large scale and innovative removal initiatives (e.g., deployment of interceptors at river mouths) are currently supporting cleanup actions worldwide on an experimental basis (Lindquist, 2016;

Zhongming et al., 2019). At a smaller scale, oil spill booms or barriers have also been adapted to aggregate riverine litter in European river basins heavily exposed to the impacts of intense human activity, facilitating the collection and the analysis of litter composition (Gasperi et al., 2014). However, the efficiency of this type of devices is still not properly understood and can be conditioned by the wind, hydrology and morphological conditions of rivers (van Emmerik and Schwarz, 2020; Andrés et al., 2021). Storms result in large flows of water and thus riverine litter fluxes to the coastal and marine environment. A well-

adapted device to storm-specific events must be considered when deciding which tools implement for a cost-effective plastic intervention strategy in the area. Further monitoring efforts are also required to account for seasonal variability on abundance



and riverine litter typology. Within the LIFE LEMA project, two videometry systems were installed at the Oria and Adour river mouths and a detection algorithm was developed to monitor litter inputs in near real time (Delpey et al., 2021; Ruiz et al., 2020b). Besides monitoring, information collected by the videometry systems can complement floating barriers collection

and sampling and advise local authorities for a quick response on riverine litter contribution to coastal area during storm events. Monitoring tools based on visual observations as RIMMEL or CrowdWater apps (González-Fernández, 2017; van Emmerik, 2020) can be also particularly helpful to build a database of riverine litter input to the SE Bay of Biscay so far remained limited or even non-existent, following a harmonized approach. Both data provided by cameras and visual observations can be crucial to evaluate the efficiency of mitigation measures as the installation of floating barriers as well as

prevention measures applied inland the river basins for a successful reduction of litter inputs into the SE Bay of Biscay.

## 6 Conclusions

The SE Bay of Biscay has been regarded as an accumulation zone for marine litter but further improve understanding of floating macrolitter behaviour originated inland is required. Research on floating marine litter and pathways at sea are

increasing but the understanding of the fate of floating macrolitter originated inland and transported through river systems is scarce and needs to be further studied. Based on HF radar current observations and wind dataset for the period 2009-2021, this contribution tries to fill this gap by providing insights on how low and highly buoyant riverine litter released by several rivers of the SE Bay of Biscay may affect the nearby regions seasonally in terms of concentration and beaching. Analysis of riverine samples collected by a floating barrier placed in the study area showed that low buoyant objects were predominant as riverine

litter although highly buoyant objects were also relevant in terms of weight. Simulations for assessing the seasonal trends of floating riverine litter transport and fate were performed with the Lagrangian model TESEO. To properly integrate the differences in litter buoyancy, simulations were parametrized with a wind drag coefficient for low and highly buoyant items. The wind drag for highly buoyant item was estimated by comparing the observed and the modelled positions of four drifters and turned out to be greater than the commonly assumed value for oil spill studies. The developed "Low-cost buoys" proved

to be suitable to provide real time trajectories of highly buoyant objects exposed to wind but drifters with different characteristics should be used in future studies for accounting the windage effect on different type of items. The transport and fate of both highly and low buoyant items released by rivers was calculated by season. Highly buoyant items rapidly beached (in less than 48 hours), particularly in summer and winter; in contrast, despite the season over two thirds of low buoyant items remained floating after one week of being released. This highlights the discrepancy between behaviour for low and highly

buoyant objects and the importance of parametrizing the windage effect in order to accurately predict riverine litter accumulation in the coastal area of the SE Bay of Biscay. Beached particles were mainly found in Gipuzkoa regardless the season and the wind drag coefficient. Overall, the less affected region was Bizkaia with the exception of Spring period for low buoyant items. Despite of the season, most of the riverine litter remained in the study area and rivers polluted the regions within the river basin or surrounding. Investigating what beaches are most likely to accumulate large quantities and the

contribution per river can provide relevant input to response operations after storm events in the short to medium term and can also support the identification of priority rivers for monitoring program, assisting in the future for an adapted intervention of riverine pollution regionally.



**7 Appendices**

**Appendix A. Floating barrier for riverine litter collection**

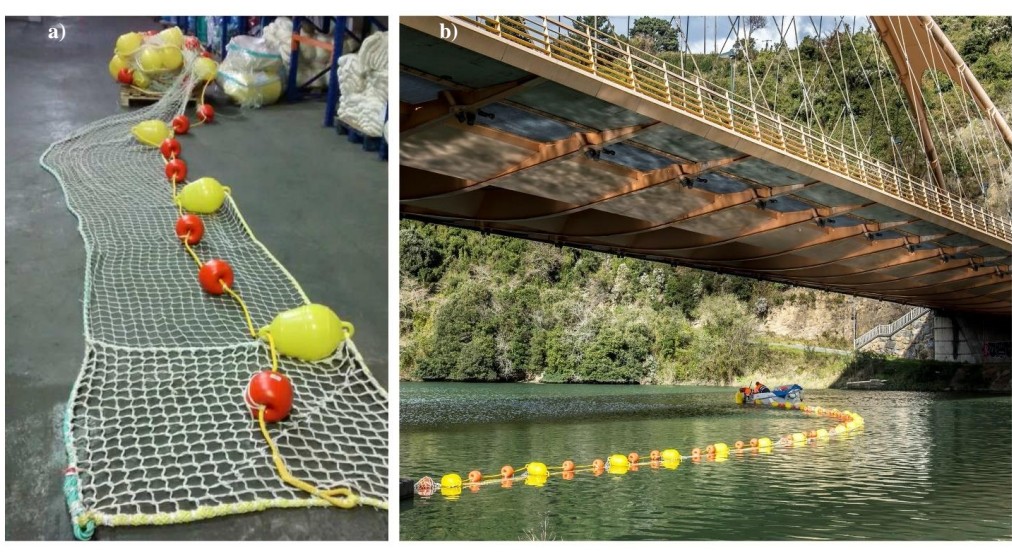

Appendix A. Floating barrier (a) and installation in Deba river (Gipuzkoa) (b)

**Appendix B. Riverine litter classification based on the exposure to wind effect**

Appendix B. Data were gathered from surveys carried out during Spring 2018 in Deba river (Gipuzkoa)

| TSG_ML General code | General name | Number of items | Weight (kg) |
|---|---|---|---|
| **Low buoyant items transported by currents** | | | |
| G1 | 4/6-pack yokes, six-pack rings | 1 | 3.3 |
| G2 | Bags | 7 | 170.7 |
| G3 | Shopping bags incl. pieces | 8 | 292.44 |
| G4 | Small plastic bags, e.g freezer bags | 4 | 50.9 |
| G5 | What remains form rip-off plastic bags | 21 | 186.31 |
| G20-G24 | Plastic caps and lids/Plastic rings | 38 | 216.39 |
| G26 | Cigarrette lighters | 1 | 9.7 |
| G27 | Cigarrette butts and filters | 1 | 0.1 |
| G30 | Crisps packets/sweet wrappers | 56 | 250.2 |
| G31 | Lolly sticks | 1 | 2.4 |
| G32 | Toys and party poppers | 2 | 97.5 |
| G36 | Fertilisers/animal feed bags | 1 | 11.5 |
| G48 | Synthetic rope | 2 | 6.7 |
| G76 | Plastic/polystyrene pieces 2.5 cm> < 50 cm | 1122 | 1788.32 |
| G77 | Plastic/polystyrene > 50 cm | 13 | 337.34 |
| G96 | Sanitary towels/panty liners/backing strips | 35 | 1099.8 |



| G100 | Medical/Pharmaceutical containers/tubes | 7 | 69.4 |
|------|------------------------------------------|---|------|
| G101 | Dog faeces bag | 2 | 106 |
| G124 | Other plastic/polystyrene items (identifiable) | 58 | 1958.5 |
| G125 | Ballons and ballon sticks | 5 | 1.1 |
| G134 | Other rubber pieces | 1 | 1.6 |
| G135 | Clothing (clothes, shoes) | 3 | 481.7 |
| G145 | Other textiles (incl. rags) | 7 | 320.5 |
| G148 | Carboard (boxes & fragments) | 3 | 85.7 |
| G156-157 | Paper & Paper fragments | 2 | 121.2 |
| G158 | Other paper items | 4 | 69.1 |
| G159 | Corks | 4 | 21.2 |
| G173 | Other (specify) | 21 | 99.3 |
| G177 | Foil wrappers, aluminium foil | 1 | 7 |
| G179 | Bottle caps, lids & pull tabs | 1 | 0 |
| | **Total** | **91.12%** | **67.95%** |
| **Highly buoyant items transported by wind and currents** | | | |
| G7 | Drink bottles <= 0.5 l | 5 | 142.6 |
| G8 | Drink bottles > 0.5 l | 3 | 91.1 |
| G9 | Cleaner bottles & containers | 2 | 105.7 |
| G10 | Food containers incl. Fast food containers | 98 | 723.9 |
| G11-12 | Cosmetics bottles & other containers (shampoo, shower gel, deodorant) | 4 | 100.3 |
| G17 | Injection gun containers | 1 | 18.3 |
| G33 | Cups and cup lids | 6 | 32.6 |
| G150-151 | Cartons/Tetrapack | 2 | 121.2 |
| G153 | Cups, food trays, food wrappers, drink containers | 4 | 69.1 |
| G174 | Aerosol/Spray cans industry | 2 | 143.2 |
| G175-176 | Bottle caps, lids & pull tabs | 2 | 5 |
| G177 | Bottles incl.Pieces | 5 | 1832.3 |
| G178 | Light bulbs | 1 | 31.7 |
| | **Total** | **8.88%** | **32.05 %** |





**Appendix C. Selected seasonal week-long periods from the HF radar (2009-2021)**

Appendix C. Periods selected between 2009 and 2021 based on the availability surface current datasets provided by the HF radar

**Winter**

|  | Period 1 | Period 2 | Period 3 | Period 4 | Period 5 | Period 6 | Period 7 | Period 8 | Period 9 | Period 10 |
|---|---|---|---|---|---|---|---|---|---|---|
| Initial date | 07-Feb-2013 08:00:00 | 09-Mar-2021 22:00:00 | 23-Jan-2009 01:00:00 | 02-Jan-2013 11:00:00 | 18-Jan-2016 17:00:00 | 02-Jan-2014 15:00:00 | 17-Feb-2017 06:00:00 | 17-Jan-2012 09:00:00 | 22-Jan-2017 17:00:00 | 12-Jan-2021 23:00:00 |
| Final date | 14-Feb-2013 07:00:00 | 16-Mar-2021 21:00:00 | 30-Jan-2009 00:00:00 | 09-Jan-2013 10:00:00 | 25-Jan-2016 16:00:00 | 09-Jan-2014 14:00:00 | 24-Feb-2017 05:00:00 | 24-Jan-2012 08:00:00 | 29-Jan-2017 16:00:00 | 19-Jan-2021 22:00:00 |

**Spring**

|  | Period 1 | Period 2 | Period 3 | Period 4 | Period 5 | Period 6 | Period 7 | Period 8 | Period 9 | Period 10 |
|---|---|---|---|---|---|---|---|---|---|---|
| Initial date | 14-Apr-2015 23:00:00 | 16-May-2012 00:00:00 | 16-Apr-2017 14:00:00 | 21-Apr-2012 08:00:00 | 05-Jun-2014 06:00:00 | 11-Apr-2021 20:00:00 | 06-May-2012 06:00:00 | 10-Apr-2015 08:00:00 | 08-May-2018 22:00:00 | 22-Apr-2016 11:00:00 |
| Final date | 21-Apr-2015 22:00:00 | 22-May-2012 23:00:00 | 23-Apr-2017 13:00:00 | 28-Apr-2012 07:00:00 | 12-Jun-2014 05:00:00 | 18-Apr-2021 19:00:00 | 13-May-2012 05:00:00 | 17-Apr-2015 07:00:00 | 15-May-2018 21:00:00 | 29-Apr-2016 10:00:00 |

**Summer**

|  | Period 1 | Period 2 | Period 3 | Period 4 | Period 5 | Period 6 | Period 7 | Period 8 | Period 9 | Period 10 |
|---|---|---|---|---|---|---|---|---|---|---|
| Initial date | 19-Aug-2017 01:00:00 | 04-Jul-2015 16:00 | 15-Aug-2016 18:00:00 | 08-Aug-2012 11:00:00 | 14-Aug-2015 00:00:00 | 08-Sep-2013 23:00:00 | 11-Sep-2017 11:00:00 | 13-Sep-2015 02:00:00 | 08-Jul-2019 4:00 | 05-Aug-2014 20:00:00 |
| Final date | 26-Aug-2017 00:00:00 | 11-Jul-2015 15:00 | 22-Aug-2016 17:00:00 | 15-Aug-2012 10:00:00 | 20-Aug-2015 23:00:00 | 15-Sep-2013 22:00:00 | 18-Sep-2017 10:00:00 | 20-Sep-2015 01:00:00 | 15-Jul-2019 3:00 | 12-Aug-2014 19:00:00 |

**Autumn**

|  | Period 1 | Period 2 | Period 3 | Period 4 | Period 5 | Period 6 | Period 7 | Period 8 | Period 9 | Period 10 |
|---|---|---|---|---|---|---|---|---|---|---|
| Initial date | 16-Oct-2014 22:00 | 17-Oct-2011 8:00 | 24-Oct-2015 11:00 | 08-Nov-2011 17:00:00 | 10-Dec-2020 10:00:00 | 06/11/2015 1:00 | 23-Nov-2015 21:00:00 | 04-Oct-2017 23:00:00 | 04-Oct-2015 20:00:00 | 23-Nov-2020 04:00:00 |
| Final date | 23-Oct-2014 21:00 | 24-Oct-2011 7:00 | 31-Oct-2015 10:00 | 15-Nov-2011 16:00:00 | 17-Dec-2020 09:00:00 | 13/11/2015 0:00 | 30-Nov-2015 20:00:00 | 11-Oct-2017 22:00:00 | 11-Oct-2015 19:00:00 | 30-Nov-2020 03:00:00 |




**Appendix D. Seasonal mean current and wind fields (2009-2021)**



Appendix D. Mean current (A) and wind fields (B) in the study area during each season for the selected periods between 2009 and 2021. The colour-bars represent the magnitude of current and wind speed. The arrows indicate the current and wind mean direction and are scaled with currents and wind speed (Data source: HFR – EuskOOS https://www.euskoos.eus/en/data/basque-ocean-meteorological-network/high-frequency-coastal-radars/ and ERA5 https://www.ecmwf.int/en/forecasts/datasets/reanalysis-datasets/era5 )



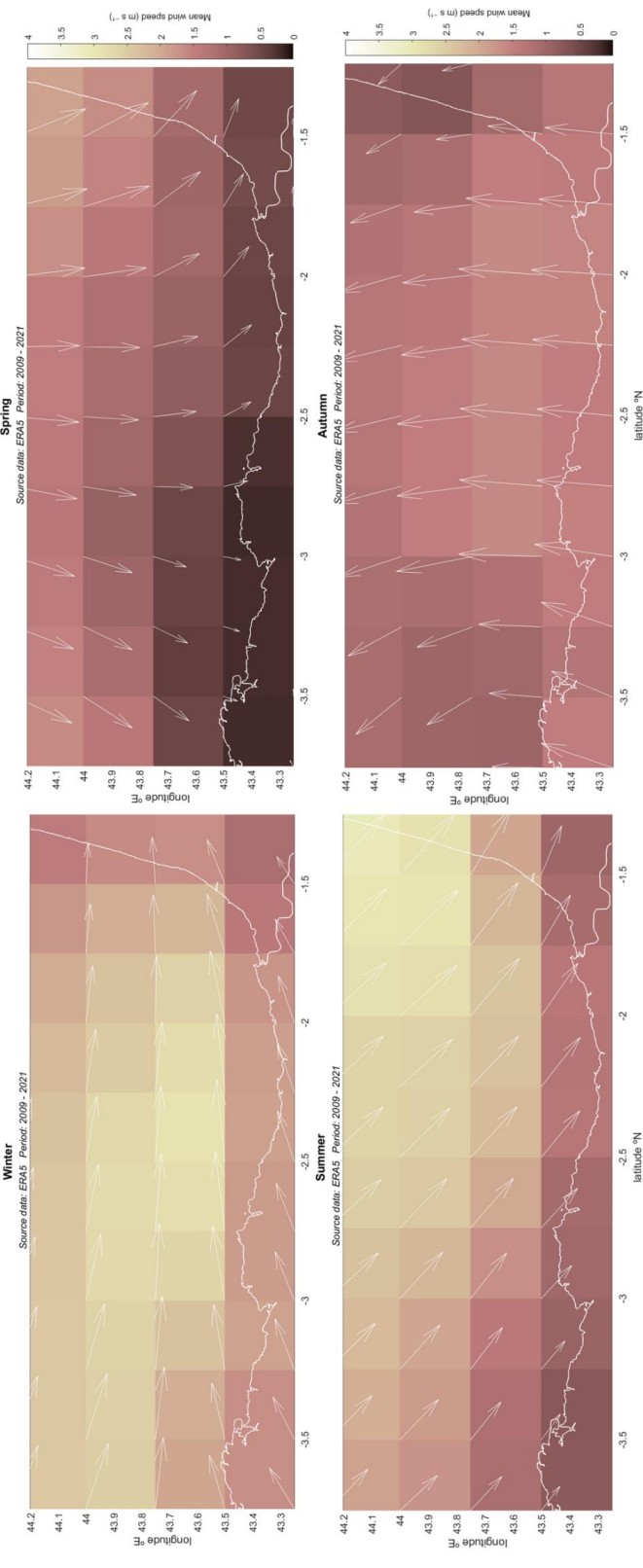



**8 Data availability**

The wind fields used for calculation in Sect. 3.5.1 and Sect. 3.5.2 are from ERA5 product
(https://cds.climate.copernicus.eu/cdsapp#!/dataset/reanalysis-era5-single-levels?tab=overview last access: 17 May 2022)
provided by ECMWF. The surface current data used for calculation in Sect. 3.5.1 and Sect. 3.5.2 are available at
https://resources.marine.copernicus.eu/product-detail/INSITU_GLO_UV_L2_REP_OBSERVATIONS_013_044/INFORMATION
and https://www.euskoos.eus/en/data/basque-ocean-meteorological-network/high-frequency-coastal-radars/ provided by CMEMS.
The litter data gathered from surveys in the riverine barrier is attached in the Appendix B. The drifter observations used for
calculation in Sect. 3.5.1 and the trajectory files obtained in Sect. 3.5.1 and Sect. 3.5.2 are available upon request. Please
contact Irene Ruiz (iruiz@azti.es).

**9 Video supplement**

Animations of the surface currents, winds and Lagrangian simulations area available for the study period 2009-2021.

**10 Author contributions**

IR: Investigation, formal analysis, visualization and writing – original draft preparation. AJ: Conceptualization, methodology,
software, writing – review & editing. OCB: Conceptualization, supervision, resources, review and editing. AR:
Conceptualization, methodology, supervision, resources, review and editing. All authors contributed to refining the manuscript
for submission. This paper is part of the PhD research of IR supervised by OCB and AR.

**11 Competing interests**

The authors declare that they have no conflict of interest.

**12 Financial support**

This research has been partially funded through the EU's LIFE Program (LIFE LEMA project, grant agreement no.
LIFE15 ENV/ES/000252) and by EU's H2020 Program (JERICO-S3 project, grant agreement No. 871153).

**13 Acknowledgements**

We are grateful to the Emergencies and Meteorology Directorate – Security department – Basque Government for public data
provision from the Basque Operational Oceanography System EuskOOS. Authors also acknowledge support from the
JERICO-S3 project, funded by the European Union's Horizon 2020 Research and Innovation Program under grant agreement
no. 871153. This study has been conducted using EU Copernicus Marine Service information. We thank Luis Ferrer for sharing
his valuable knowledge on custom-built the drifters. We thank Cristina Barreau, Antoine Bruge, Igor Granado, Théo Destang,
Alix McDaid and Jon Andonegi for their support with drifters' releasing. We thank the citizens who collected and reported
drifters' arrival to Basque and French coasts. The bathymetric metadata and Digital Terrain Model data products have been
derived from EMODnet Bathymetry Consortium (2020): EMODnet Digital Bathymetry (DTM)
(https://doi.org/10.12770/bb6a87dd-e579-4036-abe1-e649cea9881a). This is contribution number XXX of AZTI - Marine
Research (BRTA).

**14 Review statement**

This paper was edited by XXX and reviewed by XXX anonymous referees.

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
