# Peer review of "Modelling floating riverine litter in the south-eastern Bay of Biscay: a regional distribution from a seasonal perspective"

_EGUsphere, 2022_

## Author Comment (AC1)

**GENERAL COMMENTS**

Regarding the manuscript writing, a deep revision is needed. There are many sentences that are incorrect or difficult to understand. Please try to avoid very long sentences containing a lot of information. It is sometimes very complicated to understand what you are trying to point out. The vocabulary and punctuation should also be revised. About the methodology, the description of the Lagrangian model should be extended, particularly describing how does the model simulate the particles beaching. This is crucial to make a proper interpretation of the results. Also, the limitations of the model and the simulations set-up should be included in this section (instead of only the discussion). This way the reader can make a better interpretation of the results. In the results/discussion sections, you find very interesting results but a deeper analysis of some of these results and more contextualization is missing.

**Authors' response:** Thank you for your comments. We have revised and edit the manuscript to address the linguist and spelling mistakes. We have simplified and rephrased many sentences to be much more straightforward. We have extended the sub section 3.5 Particle transport model to provide a more detailed description of model (including the approach followed for beaching). Besides, we have rewritten the sub section 5.5 Model Limitations to spotlight the assumptions and simplification made on the simulation process. Finally, we have conducted a deeper analysis on the results to reinforce the sub section 3.5 Seasonal trends on floating riverine litter transport and fate.

**SPECIFIC COMMENTS**

**1.Introduction**

Lines 42-43: what do you mean with "less than a tenth"? With respect to the values given by MPW models? It is unclear what you mean here. Please rephrase.

**Authors' response:** Thank you for pointing this out. We agree with this comment. We have shortened the description of MPW models and rephrased the sentence as follows:

"Indeed, riverine litter contributions to oceans are still uncertain, and results vary depending on the input data and the model applied (Lebreton et al., 2017; Schmidt et al., 2017; Mai et al., 2020)."

Lines 45-48: This sentence is too long and difficult to understand.

**Authors' response:** Agree. We have cut it in a half for a better understanding of the sentence and now reads as follows:

"Such comprehensive data was obtained in Europe thanks to the RIMMEL project (González-Fernández and Hanke, 2017). This research concluded that between 307 and 925 million floating riverine litter items are annually transferred into the ocean, mainly through small rivers, streams and coastal run-off (González-Fernández et al., 2021)."

Line 50: what do you mean by "river waters"? If you mean that the ML remains close to the river mouth you should use this term ("river mouth").

**Authors' response:** Agree. We have rephrased the sentence as follows to make it clearer and more precise:

"Floating riverine litter can accumulate close to the river mouth or it can move long distances, reaching remote areas far from the coast."

Line 59: What do you mean by "mature"?

**Authors' response:** Thank you for this question. The research on floating litter behaviour in the coastal area is still in its early stage. Further modelling efforts and field and laboratory experiments are necessary to better understand the impact of windage on the transport of floating marine litter. Since the term "mature" can be confusing, we have rewritten the sentence to make it as clear as possible. Now, the sentence reads as follows:

"This wind effect ("windage") on floating litter behaviour has been further investigated in Lagrangian modelling studies in the open ocean (Allshouse et al., 2017; Maximenko et al., 2018; Lebreton et al., 2019; Abascal et al., 2009) when compared to the coastal area (Critchell and Lambrechts, 2016; Utenhove, 2019; Tong et al., 2021)."

Line 79: delete extra "the"

**Authors' response:** Thank you for your comment. We have deleted the extra "the" as the reviewer suggested.

Line 90: I don't fully understand why you differentiate throughout the text riverine and floating litter. Once the litter is in the sea is all marine litter. Moreover, you only simulate the ML at sea, so for your simulation experiment everything is marine little. This distinction along the text is often confusing.

**Authors' response:** Thank you for this comment. You have raised an important point here. Therefore, we have accordingly included significant changes throughout the manuscript to avoid mixing terms which can confuse the reader. We have also emphasized that the floating fraction of riverine litter simulated in this study comprises only the items that reach the open waters of the SE Bay of Biscay. For example, the following paragraph from the abstract (lines 8-12) has been modified to bring more clarity about the purpose of the study and now reads as follows:

"Although rivers contribute to the flux of litter to the coastal and marine environment, estimates of riverine litter amounts are scarce and detailed studies on floating riverine litter behaviour once items arrive to ocean are still scarce. This paper provides a comprehensive overview of the seasonal trends of floating riverine litter transport and fate in the open waters of the south-eastern Bay of Biscay based on riverine litter characterization, drifters, high-frequency radars observations and Lagrangian simulations."

Lines 94-95: "…parameterized to represent riverine litter trajectories according to their observed buoyancy." à This is not completely true. In your numerical experiment you don't use the observations to characterize the ML particles simulated. You only made a distinction between high and low buoyant particles. But the number of particles released on each simulation is always the same and with similar characteristics.

**Authors' response:** Agree. We have accordingly rephrased the sentence as follows:

"To do so, a Lagrangian model was forced by real observations from the EuskOOS HF radar and particles were parameterized to represent floating marine litter trajectories of two groups of items according to their buoyancy."

Line 100: You could include here a short description of the paper sections.

**Authors' response:** Thank you for the suggestion. We have reduced the introduction section as Reviewer#2 pointed out. Therefore, we do not have included the description to avoid loading this part of the manuscript.

**2. Study area**

Line 113: what do you mean by "self-water masses"?

**Authors' response:** Thank you for your question. By "self-water masses" we mean the oceanic water body located over the continental shelf. We have rephrased the sentence for clarity and now reads as follows:

"Over the continental shelf, the ocean circulation is marked by a seasonal variability."

Lines 115-116: "…Tidal currents in the area are quite week constrained by topography and width on the continental shelf…" à "…Tidal currents in the area are quite week, constrained by topography to the continental shelf…".

**Authors' response:** Agree. We have accordingly modified the sentence to avoid this spelling mistake. The sentence now reads as follows:

"Tidal currents are quite weak  constrained by the topography and the width of the continental shelf (Lavin et al., 2006; González et al., 2007; Karagiorgos et al., 2020)."

Lines 118-120: This sentence is too long. Rephrase please.

**Authors' response:** Agree. We have cut it in a half and rephrased the sentence for a more understandable description of the circulation. The paragraph now reads as follows:

"In winter, the prevailing SW winds causes an E to N flow from the Spanish coasts towards the French coasts. The moderate to strong NW winds occurring in spring and summer induce S and SW surface currents circulation accompanied by a greater variability (Solabarrieta et al., 2015)."

Line 121: Achieving à reaching

**Authors' response:** Agree. We have modified the sentence for using the adequate term suggested by the reviewer and now reads as follows:

"In winter, westerly winds in the Basque coast reinforce the slope current (named "Iberian Poleward Current" (IPC)), a warm and saline intrusion trapped within the 50 km of the shelf edge, reaching its greatest velocities (up to 70 cm s-1) during this season."

Line 127: Very strange to cite a figure from other paper. Include the figure number please.

**Authors' response:** Agree. This specific point has been also raised by Reviewer#2 so we have deleted the sentence and reference.

Lines 129-136: Here you have to be more specific in the description of the results you are citing. Most of these studies are based on Lagrangian simulation of ML particles, many of them using numerical models for the current fields, others using HF radar data. Some of them include windage, with different parameterizations, others don't, etc… You should specify the details of the estimations you are citing and also try to avoid the word "observed", since these results are mostly based on simulations. I would also include here a short summary of the most important sources of uncertainty found by the authors in their different approaches. I think is important to contextualize the results of the study and the limitations of the state-of-the-art ML modelling.

**Authors' response:** Thank you for pointing this out. We agree with this comment. We have dealt with your suggestions by avoiding the term "observed" and amending this section to include a brief description of the cited works. The paragraph added reads as follows:

"First global modelling studies coupling ocean circulation and Lagrangian particle tracking models reported that the SE Bay of Biscay is a hotspot for floating marine litter (Lebreton et al., 2012; van Sebille et al., 2012). A recent Lagrangian modelling study combining measured and predicted surface currents by the HF radar and the IBI Copernicus model revealed that floating marine litter circulation in the SE Bay of Biscay is marked by a high seasonal variability. Results showed a higher retention during spring and summer and a northward dispersion along the French coast during autumn and winter (Declerck et al., 2019; Rubio et al., 2020). Surface currents derived from Regional Ocean Modelling System (ROMS) and a particle-tracking model were combined by Pereiro et al., 2019 to track the numerical drifters representing floating marine litter in the Bay of Biscay. In this study, longer residence times and higher concentrations were observed in the SE Bay of Biscay when compared to north-western Iberian coastal waters, particularly in winter. Rodríguez-Díaz et al., 2020 showed from numerical simulations run using HYCOM model that floating litter items with high windage (Cd=3%-5%) tend to accumulate in nearshore areas of the Bay of Biscay or end up beached. These trend is consistent with recent numerical simulations combining surface currents from the operational Iberian Biscay Irish System (IBI) and the numerical model TESEO that also

revealed the highly buoyant items (Cd=4%) rapidly beach in the SE Bay of Biscay, mainly during spring and summer (Ruiz et al., 2022a).

We have also underlined the particular limitations for accurate model the transport and distribution of floating marine litter in the study area by including the following sentence:

"However, research on floating marine litter behaviour in the SE Bay of Biscay is still in its early stage. Further experiments are needed to fully understand the role of windage, waves and tides in the complex 3D circulation patterns governing coastal accumulation."

**3. METHODS AND DATA**

**3.2 Drifter observations**

Lines 163-164: This is very interesting. Could you provide a little more information about the batteries and its duration? According to the table some of the buoys worked only for a few days and others for more than 2 weeks. Why this difference? Did you recovered the buoys or were lost?

**Authors' response:** Thank you for your comment and for raising these questions. The drifting buoys were powered by 4 AAA cells. For none of the experiments, the SPOT Trace showed that the battery power reserves were low. However, Buoy C and D stopped emitting without warning for reasons yet unknown. One reason may be that the GPS detached from the bottom of the container and consequently the signal was lost. These SPOTs Trace have been reused in subsequent campaigns and their performance was good.

We have amended this section to provide further details on the buoys performance and their recovered. The new paragraph reads as follows:

"Four satellite drifting buoys (herein after 'low-cost buoys') were built by the authors and deployed one-by-one in the river mouths of Deba (Buoy A), Oria (Buoy B), and Adour (Buoy C and D) between April 2018 and November 2018 (Fig 1, Table 1). The 'low-cost buoys' provided positioning every 5 minutes using satellite technology. 'Low-cost buoys' were 9 cm in height, 9.5 cm in float diameter and weighed approximately 200 g (Fig 2). A GPS (SPOT Trace device) powered by 4 AAA cells was placed in the bottom of a high-density polyethylene plastic container sealed to guarantee water tightness. Almost 2/3 of the buoy floated above the water surface thus preventing any satellite signal losses. Buoys A and D transmitted their positions on an ongoing basis until their landing. Buoys B and C stopped emitting while they were drifting. In all cases, battery lifetime was enough for an adequate performance of the buoys. Once on land, citizens collected the buoys and reported their corresponding location."

**3.3 HF radar current observations and wind data**

Lines 179-180: This sentence is too long. Rephrase please.

**Authors' response:** Thank you for the suggestion. We have revised this sub section to provide a more detailed description on the quality control procedures, so we have deleted this sentence. The new paragraph reads as follows:

"85 OMA modes, built setting a minimum spatial scale of 20 km and applied to periods with data from the two antennas, were used to provide the maximum spatiotemporal continuity in the HFR current fields, which is a prerequisite to performing accurate Lagrangian simulations. The application of OMA methodology has been validated for the Lagrangian assessment of coastal ocean dynamics in the study area by Hernandez-Carrasco et al. (2018). HF radar velocities were quality controlled using procedures based on velocity and variance thresholds, signal-to-noise ratios, and radial and total coverage, following standard recommendations (Mantovani et al., 2020). Data subsets were built for the Lagrangian simulations avoiding periods with temporal gaps (still present in case of failure of one or the two antennas) of more than a few hours."

Line 184: The resolution is 30 km or 0.3°x0.3°. Both are similar but not exactly the same. Giving two different values is confusing.

**Authors' response:** Thank you for pointing this out. We agree with this comment. Therefore, we have accordingly deleted the replicated resolution description and the new sentence reads as follows:

"ERA5 atmospheric database covers the Earth on a 30 km horizontal grid using 137 vertical levels from the surface up to a height of 80 km and provides estimates of a large number of atmospheric, land and oceanic climate variables, currently from 1979 to within 3 months of real time."

Lines 185-186: the weekly periods are first mentioned here. Either described them or indicate the section/table where you describe them below.

**Authors' response:** Agree. We have updated the manuscript by citing the corresponding table that describes the periods (Appendix C). Now the sentence reads as follows:

"Both HF radar current observations and wind data cover the drifter's emission periods and the selected week-long periods between 2009 and 2021 for riverine litter simulations (see Appendix C for the selected periods)."

**3.5 Particle transport model**

Here a much more detailed description of the Lagrangian model is missing. This is crucial to understand the accuracy of the results.

**Authors' response:** Thank you for pointing this out. We agree with this comment. We have therefore dealt with your suggestion by amending this subsection to include a more extended and detailed description of the particle transport model. The specific questions are addressed below in a point-by-point manner.

Some of the missing information is:

1. How does the model solve the movement of the water parcels?

**Authors' response:** The transport module of the TESEO particle-tracking model allows for simulations of passive drifters driven by surface currents, wind and turbulent diffusion. The trajectories of the drifters are calculated using the following equation:

$$\frac{d\vec{x_i}}{dt} = \vec{u_a}(\vec{x_i},t) + \vec{u_d}(\vec{x_i},t)$$

where $\vec{u_a}$ and $\vec{u_d}$ are the advective velocity and diffusive velocity, respectively, for the $\vec{x_i}$ point and t time. The advective velocity is calculated as the lineal combination of the wind and currents according to:

$$\vec{u_a} = \vec{u_c} + C_d\vec{u_w}$$

where $\vec{u_c}$ is the surface current velocity, $\vec{u_w}$ is the wind velocity at 10m over the sea surface and Cd is the wind drag coefficient.

2. Does it include horizontal diffusion? If so, how it is implemented? Random walk? I understand from table 2 that turbulent diffusion is included, but not explained.

**Authors' response:** The turbulent diffusive velocity is obtained using Monte Carlo sampling in the range of velocities $[-\vec{u_d}, \vec{u_d}]$ which are assumed to be proportional to the diffusion coefficients (Hunter et al., 1993; Maier-Reimer and Sündermann, 1982). For each timestep Δt, the velocity fluctuation is defined as:

$$|\vec{u_d}| = \sqrt{\frac{6D}{\Delta t}}$$

where D is the diffusion coefficient, whose value is 1 m²/s in accordance to previously modelling work for floating marine litter (Pereiro et al., 2019; Ruiz et al., 2022).

3.  Since the resolution of the HF radar and the wind data are different, I understand that the wind data is interpolated to the HF radar grid, am I right?

**Authors' response:** Both wind data and surface currents were interpolated at the particle position for integrating the trajectories.

4.  How is the wind drag coefficient implemented in the movement equations?

**Authors' response:** This specific point has been addressed previously for question 1.

5.  How do you define when a particle is beached? Does the Lagrangian model includes a beaching algorithm? How does it work? This is particularly important since some of your more relevant results are related to the beaching process. A detailed description of how the model considers a particle beached is crucial to understand your results.

**Authors' response:** Beaching along the coast was implemented by a simple approach: if the particle reaches the shoreline, it is identified as beached and it is removed from the computational process.

**3.5.2 Lagrangian seasonal simulation of riverine litter items**

Line 224: Please indicate the total number of simulations (40, if I'm not wrong).

**Authors' response:** Thank you for your comment. We have dealt with your suggestion by including the total number of simulations and their distribution according to the windage parametrization. We have rephrased the paragraph as follows to make it clearer:

"In total, 80 simulations (40 for Cd=0% and 40 for Cd=4%) were run for 7 days. For each simulation, 4,000 particles were released in 8 rivers (500 per river) (Table 2)."

Lines 228-227: This is another key issue that you should underline and also take into account in the discussion section when comparing the results with previous works and observations. You are releasing the ML particles 2.5 miles from the coast. As the authors know for sure, there are numerous coastal processes that traps the ML in coastal areas, especially if the wind drag is taken into account. Therefore, your results are only valid for the fraction of ML coming from the rivers that leaves the coastal area and reaches open sea. In addition, you are not making any difference between rivers or seasons. You are considering that all the rivers have the same ML input, and that this input is constant along the whole year. Therefore, the spatial distribution and seasonality that you obtain only depends on the river mouth position and the variability of the HF radar current field and the ERA5 wind field. In summary, you are considering the 8 rivers as constant ML input sources on open sea. I think that it is important to state very clearly these assumptions, and the limitations that imply, in order to make a proper interpretation of your results. In section 5.3 and 5.4 you address some of these issues, but in my opinion is important to clarify them here, before presenting the results. This way the reader is aware of the model/simulations limitation and can make a better interpretation.

**Authors' response:** Thank you for this comment. You have raised important points here. Therefore, we have accordingly included changes throughout this sub section to clarify them. Firstly, we have emphasized that the particles released in the simulations represent the floating fraction of riverine litter that leaves the coastal area of the SE Bay of Biscay:

"Seasonal simulations were run for low and highly buoyant items to assess the seasonal differences on the transport and fate of riverine litter reaching the open waters of the SE Bay of Biscay. Particles were released around 2.5 nautical miles off the coastline due to the complexity in resolving small-scale processes of floating litter behaviour in and close to the river mouths."

We have also spotlighted that no seasonal differences on river flow and between rivers were taken into account in the study. The new sentence reads as follows:

"In total, 80 simulations (40 for Cd=0% and 40 for Cd=4%) were run for 7 days. For each simulation, 4,000 particles were released in 8 rivers (500 per river) assuming that river discharges are equal despite the seasonal variations and the morphological differences between rivers (Table 2)."

**4. RESULTS**

**4.1 Riverine litter characterization**

I think these results are very interesting. There is a lack of information on the ML sources in general, and on rivers in particular. This kind of experiments are very useful to start filling these knowledge gaps.

**Authors' response:** Thank you for this nice and valuable comment. Regarding the freshwater environment, only a few research efforts have been dedicated to study riverine litter so we hope this characterization would provide helpful information to other colleagues working on the issue.

**4.2 Wind draft coefficient for drifting buoys**

Lines 259-260: I don't understand what you mean by "spread out over the rivers inside the HF radar coverage area". Please clarify this sentence.

**Authors' response:** Thank you for pointing this out. There was a writing mistake which turned it into an unintelligible sentence. We have rephrased the sentence and now reads as follows:

"Total distances covered by drifting buoys ranged from 62 km to 118 km (Table 1) and they all scattered over the HF radar coverage area. Buoys provided their position data over 385 h before beached on Landes and Gipuzkoa shorelines."

**4.3 Seasonal trends on floating riverine litter transport and fate**

Lines 277-278: Very interesting result but depends on the beaching parameterization.

**Authors' response:** Thank you for your comment. We have updated the manuscript by improving the sub section 3.5 Particle Transport Model to provide a broader description on how the process of beaching is implemented in the simulations.

Line 281: Which specific characteristics of the forcing are you considering in this assumption? In my opinion, there are quite different behaviors of the particles depending on the location of the river mouths. For instance, in the Urumea river almost all particles with Cd = 0% remain in the water after the week period, while for the Deba river the reduction of particles is much higher (~200 less on water). Also, there is a clear seasonality, rivers that "lost" more particles in summer and winter are different. A deeper and clearer analysis of this results is missing.

**Authors' response:** Thank you for your comments and for arising these questions. We have rewritten the analysis of the temporal evolution of the particles to make a deeper and a more comprehensive description of the results. The new paragraph now reads:

"Overall, the average of floating particles parametrized with Cd=0% was higher when comparing to Cd=4% (Fig 9). Floating particles released in French rivers and parametrized with Cd=0% were less abundant during summer, though this trend was reversed in autumn. For Cd=0%, the number of floating particles released in Bidasoa river during summer were the least abundant after one week of simulation (<200 particles on average). The vast majority of particles released in Urumea river during winter were floating in the study area by the end of the simulations (479 particles on average). Particles parametrized with Cd=4% beached faster during the first 48 hours, mainly in summer and for those particles released in the French rivers. During this season, the average number of particles floating in the study area by the end of the simulation ranged between 0 and 250. Similar trends were observed within the same season between rivers, probably influenced by the vicinity of rivers and the spatiotemporal resolution of forcings."

Line 281-283: "When... ... simulations". This sentence is too long and a bit confusing. Please rephrase to be more clear.

**Authors' response:** Agree. We have deleted the sentence to provide more understandable information to the reader.

Lines 283-286: There is a clear seasonal variability in the beaching regions, particularly for particles not affected by winds. This variability can be only linked to the current field variability. You could mention this here or in the discussion section.

**Authors' response:** Thank you for this comment. You have raised an important point here. Therefore, we have accordingly included the following sentence to highlight that the current field variability shape the distribution of beached particles parametrized with Cd=0%:

"Beached particles parametrized with Cd=0% experienced more seasonal variations derived from the surface current circulation patterns within the SE Bay of Biscay."

In general, a little more detail in the analysis of figures 9 and 10 is missing. I think they represent very relevant results of the study and a more thorough description would be adequate.

**Authors' response:** Agree. Thank you for this suggestion. We have updated the manuscript by improving this sub section with a broader description of these two figures. The new paragraph for describing them read as follow:

"Overall, the average of floating particles parametrized with Cd=0% was higher when comparing to Cd=4% (Fig 9). Floating particles released in French rivers and parametrized with Cd=0% were less abundant during summer, though this trend was reversed in autumn. For Cd=0%, the number of floating particles released in Bidasoa river during summer were the least abundant after one week of simulation (<200 particles on average). The vast majority of particles released in Urumea river during winter were floating in the study area by the end of the  simulations (479 particles on average). Particles parametrized with Cd=4% beached faster during the first 48 hours, mainly in summer and for those particles released in the French rivers. During this season,  the average number of particles floating  in the study area  by the end of the simulation ranged between 0 and 250. Similar trends were observed within the same season between rivers, probably influenced by the vicinity of rivers and the spatiotemporal resolution of forcings. Over 40% of the total particles parametrized with Cd=4% and almost 12% of parametrized with Cd=0% beached in Gipuzkoa (Fig 10). During spring, almost 60% of beached particles parametrized with Cd=0% reached Bizkaia. For Cd=0%, particles released during summer in the rivers located in the western area of Gipuzkoa drifted longer distances and reached Landes coastline. This trend changed during winter, when the vast majority of particles released in Gipuzkoa rivers beached mainly in Gipuzkoa and Bizkaia. Beached particles parametrized with Cd=0% experienced more seasonal variations derived from the surface current circulation patterns within the SE Bay of Biscay. For Cd=4%, particles beached in Gipuzkoa ranged between 51%  in spring and 38% in  winter and Bizkaia was the less affected region despite the season. Overall, all regions were highly affected by rivers within or nearby the region itself."

Figure 9: it is very difficult to distinguish the lines corresponding to each river. Please choose clearly different colors for each one.

**Authors' response:** Thank you for the suggestion. Colored lines for rivers in Figure 9 are consistent with those selected for the nodes in Figure 10. We believe that preserve selected colors can help make figures 9 and 10 easier to follow. Nevertheless, we have changed the line style properties, in particular, the marker symbol, to better distinguish them.

Figure 10: This figure is very interesting and informative. Please indicate in the figure the region to which each river belongs. Also, I think that if you put Bizkaia above Gipuzkoa, so the regions are ordered counter-clockwise (from W to NE), it would be easier to understand the particles transfers from one region to another.

**Authors' response:** Thank you for your comment and your suggestions. Agree. We have accordingly modified the figure to include the recommended changes.

**5. DISCUSSION**

**5.1 Riverine litter composition**

Lines 309-310: I don't understand the meaning of this first sentence, please rephrase it.

**Authors' response:** Agree. We have accordingly modified the sentence for a clearer understanding. The sentence reads as follows:

"An artisanal net placed at the mouth of Deba river enable sampling riverine litter in the study area during Spring 2018."

Lines 323-324: If you find higher percentage of large pieces (2.5-50 cm) of polystyrene, doesn't it mean that the degradation is lower (not higher) than in the Black Sea or the Mediterranean?

**Authors' response:** Thank you for arising this question. The percentage of large items fragmented into unidentifiable pieces sized between 2.5 and 50 cm in Deba river is higher than the Black or the Mediterranean Sea. The rate of fragmentation depends on the environmental conditions and the type of material (Chamas et al., 2020; Woods et al., 2021) so these results can be motivated the by differences on the type of material of litter items and the weather and climate differences between river basins. We have modified the paragraph for a more detailed explanation and now reads as follows:

"Despite the morphology and hydrological differences, plastic was the predominant material in Deba river, as in Siene (Gasperi et al., 2014), Danube (Lechner et al., 2014) or Rhine River (van der Wal et al., 2015). Plastic/polystyrene pieces between 2.5 cm and 50 cm (71.2%) top the list in terms of number of items and their abundance was slightly higher when compared to North-East Atlantic rivers (54.53%) (Bruge et al., 2018; Gonzalez-Fernandez et al., 2018). Lower abundances were observed in the Mediterranean (25.01%) and the Black Sea (13.74%). Riverine litter items trapped on vegetation or deposited on the riverbank can be degraded by weather conditions (rain, wind, etc.) favouring the fragmentation in plastic pieces before their arrival to the coastal and marine environment (Chamas et al., 2020). The fragmentation can be also influenced by the material and the shape of the litter items (Woods et al., 2021). Differences on Plastic/polystyrene pieces between 2.5 cm and 50 cm abundances can be attributed to a faster fragmentation due to the variations on weather conditions between river basins. However, more detailed analyses on the physical characteristics of litter items (i.e., polymer type) are necessary to fully assess their impact on the occurrence of fragmented plastic pieces."

The size of the sampling net grid is 6 cm. Meaning that items smaller than this size will pass through the net. That is probably why you find so few bottle caps or cigarette butts, which are very common. Do you have any estimation of the amount of items between 2.5 a 6 cm that you could have missed (maybe observations at sea near the Deba's mouth)?

**Authors' response:** We agree with the reviewer that macrolitter items sized between 2.5 and 6 cm were not addressed in this study due to the mesh size. Unfortunately, neither visual observations of floating riverine litter nor beach litter campaigns were performed in the rivermouth during this period. We will take up this suggestion for those future sampling actions that imply the use of this net in order to acquire more precise data.

**5.2 Wind drag estimation**

As a suggestion, many of the considerations about the suitability and accuracy of the low cost drifters would be more useful in the methodology section.

**Authors' response:** Thank you for pointing this out. We agree with this comment. We have dealt with your suggestion by amending the Section 3.2 Drifters Observations to include the brief description on the multiple uses of the Spot Trace devices. The paragraph added reads as follows:

"They were chosen because of their capability to ensure a reasonable balance between an accurate signal emission and their purchase and communication fees. SPOT Trace devices have been used over the past few years in coastal and open ocean applications in a wide range of studies. Studies range for calibrating HF Radars (Martínez Fernández et al., 2021), tracking drifting objects as icebergs (Carlson et al., 2020), pelagic Sargassum (Putman et al., 2020; Van Sebille et al., 2021) or fishing vessels (Widyatmoko et al., 2021; Hoenner et al., 2022) to search and rescue training(Russell, 2017) (Russell, 2017) and oil spill and litter monitoring (Novelli et al., 2018; Meyerjürgens et al., 2019)."

I miss a comparison with previous studies on wind drag coefficient for Marine Litter, rather than using results for oil spill or algae. For instance, the work of Pereiro et al (2018) and references therein (https://doi.org/10.1080/1755876X.2018.1470892). Critchell et al. (2015) (https://doi.org/10.1016/j.ecss.2015.10.018) or Critchell and Lambrechts (2016) (https://doi.org/10.1016/j.ecss.2016.01.036).

**Authors' response**: Agree. We have accordingly updated the manuscript by including the estimated range of wind drag coefficient provided in other studies. The new paragraph reads as follows:

"One of the largest uncertainties for predicting floating litter behaviour is the proper quantification of a wind drag coefficient. Wind drag estimations conducted so far for floating litter items range between 0% and 6% (Ko et al., 2020; Critchell and Lambrechts, 2016; Neumann et al., 2014) with an upper limit of 10% (Yoon et al., 2010). However, only few of them have been validated using observational data (Maximenko et al., 2018; Callies et al., 2017)."

We have also included the research performed by Pereiro et al (2018) to allow for comparison of windage parametrization for floating litter items within the Bay of Biscay. The new sentence reads as follows:

"This value can be consistent with the estimations of the partially emerged Physalia physalis for the Bay of Biscay (Ferrer and Pastor, 2017) but it is almost three times higher than the maximum wind drag coefficient reported in the area by Pereiro et al., 2018. This can be explained by the fact that buoys used in the experiment remained submerged beneath the sea surface and were less exposed to wind effect."

**5.3 Seasonal riverine litter distribution by region**

Line 371: delete "but".

**Authors' response:** Agree. Thanks for your remarks about the linguistic and spelling mistakes. We have deleted it.

Lines 378-380: the last sentence is too long, please consider rephrasing.

**Authors' response:** Agree. We have rephrased the sentence to make it shortener. Now reads as follows:

"The pathways and fate of low buoyant items reflect the seasonal surface water circulation patterns in the SE Bay of Biscay. Results are in line with findings provided by (Declerck et al., 2019) who pinpointed a higher coastal retention in the area during spring and summer."

**5.4 Rivers as key vectors of riverine litter**

Lines 384-385: Indeed, what you are showing in this study is the impact of the river mouths as a constant source of ML in the ocean. All the variability described depends only on the HF radar current filed and the ERA5 wind filed (for those particles affectted by wind).

**Authors' response:** Agree. To avoid more uncertainties on the results, we decided not to include in this study i.e., the effect of river flow or the bidirectional tidal flow on the net transport of floating litter from the rivers into the SE Bay of Biscay. However, well-documented physical processes which variability may impact on the occurrence on floating riverine and marine litter in the area should be considered on future modelling approaches.

Lines 387-388: I think in your case the socio-economic factors are quite homogeneous in your area of study.

**Authors' response:** Thank you for your comment. We would like to stress that the differences on population density between regions located in the study area can affect the rates of riverine litter discharge. Accordingly, we have rephrased the sentence as follows:

*"Other drivers as the land use or population density can be a determining factor on the amount of mismanaged litter that could contribute to riverine litter fluxes (Schmidt et al., 2017; Schuyler et al., 2021)."*

We have also included a short description of the population density in the Introduction section:

*"The mean annual river discharge varies widely between rivers - 3.71 m3/s (Oiartzun) to 350 m3/s (Adour) (Sheppard, 2018) and the population density differs between the Spanish and French border – 44.8 inhabitants/km2 (Landes) to 303.7 inhabitants/km2 (Basque Country) -(Eurostat, 2019)".*

Line 394: What do you mean by "dominant number of rivers"

**Authors' response:** Thank you for arising this question. Rivers in Gipuzkoa outnumber by far Landes and Pyrénées-Atlantiques. We have rephrased the sentence to make it clearer and now reads as follows:

*"Rivers in the study area are mainly located in Gipuzkoa which favours the accumulation of floating litter in this region regardless the season."*

**5.5 Model limitations**

This is key in the processes that you are describing along the whole paper. I would put this whole section in the introduction or the methodology section. Together with a detailed explanation on how your model simulate the beaching process. In this section I would include an estimation (or at least a description) of the uncertainty specifically related with your model. As I mentioned in my comments for section 3.5.2, the model and simulations set-up limitations that are previously known could be included in that section for clarity (also for the beaching algorithm). Here I would comment the impact of those limitations on the results.

**Authors' response:** Thank you for your comment. As the reviewer suggested, we have rewritten the sub section 3.5 *Particle transport model* to give a detailed description of the model. This improvement in the manuscript supports the discussion that we have performed on the model limitations and the uncertainty of the results, including the difficulties to model the beaching processes and the need to improve it to advance in the modelling of floating litter. The new sub section now reads as follows:

"The interaction between floating litter and the shoreline is highly complex and relies in many processes including waves and tides. Indeed, waves and tides can constrain coastal accumulation since they can resuspend and transport litter back into the ocean (Brennan et al., 2018; Compa et al., 2022). The geomorphology can also affect the retention of litter washing ashore. Sandy beaches tend to be more efficient at trapping and accumulating litter than rocky areas, which favor litter fragmentation (Robbe et al., 2021; Weideman et al., 2020). How these processes contribute to the actual beaching is unknown and they cannot be resolved yet at a suitable resolution (Melvin et al., 2021). In this study, particles were released in open waters and once they reached the shoreline, they were classified as beached. The tidal effect and the wave-induced Stokes drift were not accounted for to avoid introducing more uncertainties. However, further field and laboratory experiments to better understand on how these processes influence floating litter behaviour in the coastline is recommend. It is also important to consider for future research exploring the effect of the type of shoreline on coastal accumulation. In this study, a constant diffusion coefficient of 1 m$^2$/s was considered as a pragmatic choice based on previously modelling work for floating marine litter. However, more field measurements are necessary to accurately assess the influence of the diffusion process on the transport of floating marine litter."

**5.6 Riverine litter collection and monitoring by a floating barrier**

This is very interesting, but I don't see how is related with the results of your study. I suggest to summarize and include it in the introduction section or to clearly point out the relation with your results.

**Authors' response:** We appreciate the comment and the suggestion. We agree with the reviewer that this sub section seems more disconnected from the discussion. We have accordingly removed it to make a clearer and much straightforward discussion.

**6. CONCLUSIONS**

Lines 441-443: Actually, since you don't use real data of the amount of ML transported by the rivers, you are not analyzing the input of inland ML, you are estimating the fate of the ML once it reaches open sea.

**Authors' response:** Agree. **W**e have accordingly rephrased the sentence to define more clearly the scope of the study and the implications on the results. The new sentence reads as follows:

"The SE Bay of Biscay has been described by global and regional models as an accumulation zone for floating marine litter. However, detailed studies on floating riverine litter behaviour once items arrive to open waters are still scarce."

Lines 448-449: This comparison should be made with other works estimating wind drag coefficient. In the literature this coefficient ranges between 2-1.5%, so in the range of your estimation. This should be further discussed.

**Authors' response:** Thank you for your comment. As the reviewer suggested, the comparison of windage results with previous studies has been discussed in sub section 5.2 *Wind drag estimation*.

Line 449-451: "The developed… …Type of items" à This is a very interesting result.
Congratulations.

**Authors' response:** We appreciate the comment. We have splitted into two sentences to make it shortener. Now reads as follows:

"The developed "Low-cost buoys" proved to be suitable to provide real time trajectories of highly buoyant objects exposed to wind. However, drifters with different characteristics should be used in future studies for accounting the windage effect on different type of items."

Finally, I wonder why the authors didn't combine the information obtained in the sampling of the Deba river with the numerical results. You estimate that around 68% of the riverine litter collected were low buoyancy items, while the rest 32% were high buoyancy items. Even if you keep the same number of particles in your simulations for both type of items, you could give an estimation based on the observations of the fraction of each type expected to reach the coast. For instance, according to figure 8, in winter 95% of high buoyant particles reach the beach, while for the low buoyant only 25% are beached. This mean that if both type of particles are considered (keeping the fractions observed in the Deba), only 47% of the particles would reach the beaches, 30% would be high buoyant and 17% low buoyant.

**Authors' response:** Thank you for this valuable comment. You have raised an important point here. We decided not to apply the estimated highly and low buoyant fractions to beached particles since sampling was conducted only during spring. We wanted to avoid adding further uncertainty over the results. However, we will take up this suggestion for those future research and forthcoming sampling in Deba river.

---

## Author Comment (AC2)

**General comments**

The manuscript needs comprehensive language editing. There are a lot of spelling mistakes, and many sentences are unclear to me. A thorough language editing for the manuscript is necessary to publish this study in Ocean Science.

Authors' response: Thank you for pointing this out. We agree with the reviewer's assessment. Accordingly, we have revised the spelling and grammatical errors pointed out by Reviewer#2 but also by the Reviewer#1 throughout the manuscript. Besides, we have rephrased and splitted many sentences to make then shortener and consequently provide more understandable information to the reader.

**1.Introduction**

The Introduction should be shortened. There are reiterative sentences and sections which are disconnected. Furthermore, technical details of the radar data should be moved to the methods section. References to webpages should be deleted as they just load the text.

Authors' response: Agree. We have dealt with your suggestions by reducing this section in order to present an introduction as clear as possible. We have deleted when not necessary and rephrasing many sentences, including the references to webpages. We have also moved the description on the technical aspects of the HF radar to the section 3 Methods and data.

**2.Windage**

The method used to calculate the wind slip of the particles is questionable. The referenced numerical studies do not simply add different windage values and estimate the distance of the trajectories. Please go more in-depth here and use an appropriate method to compare your numerical trajectories with those of the drifters.

Authors' response: Thank you for the suggestion. We agree with the reviewer that the approximation to the particle's behavior due to windage can be very complex since it depends on different parameters, from the shape and buoyancy of the objects to small scale processes in the air-sea interface. The fine tune of the Lagrangian model for the wind slip is out of the scope of this paper, where we focus on the submesoscale to mesoscale transport of particles in the coastal areas and how considering a simple windage approximation can be key for more accurate simulations.

Furthermore, as I understand it correctly, the particles were re-initialized every 4 hours on the drifter trajectories. This may neglect submesoscale processes that significantly affect the dispersion and distribution of floating objects in the ocean. The effects of tides may be underestimated, which of course, also play an essential role in the propagation and dispersion of particles in the Bay of Biscay. Please strengthen the study in this regard.

Authors' response: Thank you for your comment. New particles are released along the observed trajectories every 4-h but run during 24 hours in the simulations for wind drag estimation. However, for the study of seasonal scenarios of transport the particles are advected for 1 week, which provides integration of submesoscale and high frequency processes observed by the HF radar (mainly tides and eventually inertial oscillations)

**3. HF radar current observations and wind data**

The methodology of how the HF data is extracted and assimilated with the wind observations is, in my view, unclearly described. How are these data products incorporated on a uniform grid for further analysis? In addition, lines 178-180 indicate that the data extraction is questionable. Please clarify precisely how you extracted the data and what criteria were used for the quality check.

Authors' response: Thank you for your comment and for arising this question. The methodology for the processing and ingestion of the HF radar data is now improved. The new paragraph reads as follows:

"Surface velocity current fields were obtained from the EuskOOS HF radar station composed by two antennas located at Matxitxako and Higer Capes and covering the SE Bay of Biscay covering since 2009 a range up to 150 km from the coast. The EuskOOS HF radar is part of JERICO-RI (https://www.jerico-ri.eu/) and it is operated following JERICO-S3 project best practices, standards, and recommendations (see (Solabarrieta et al., 2016; Rubio et al., 2018) for details). Data consist of hourly current fields with a 5 km spatial resolution obtained from using the gap-filling OMA methodology (Kaplan and Lekien, 2007; Solabarrieta et al., 2021). "85 OMA modes, built setting a minimum spatial scale of 20 km and applied to periods with data from the two antennas, were used to provide the maximum spatiotemporal continuity in the HFR current fields, which is a prerequisite to performing accurate Lagrangian simulations. The application of OMA methodology has been validated for the Lagrangian assessment of coastal ocean dynamics in the study area by Hernandez-Carrasco et al. (2018). HF radar velocities were quality controlled using procedures based on velocity and variance thresholds, signal-to-noise ratios, and radial and total coverage, following standard recommendations (Mantovani et al., 2020). Data subsets were built for the Lagrangian simulations avoiding periods with temporal gaps (still present in case of failure of one or the two antennas) of more than a few hours."

We have also included a more detailed explanation on the interpolation of the HF radar and wind data in the model:

"Simulations were forced by HF radar surface current velocity and wind data and interpolated at the particle position for integrating the trajectories. Beaching along the coast was implemented by a simple approach: if the particle reaches the shoreline it is identified as beached and it is removed from the computational process."

**4. Particle transport model**

This paragraph does not describe the particle tracking module. The information given here is repetitive and only explains what the intent is for the particle simulations. Please describe exactly which way particle tracking was used. Are concepts for horizontal diffusion included and what scheme is used to move the particles forward in the module? It is not sufficient to cite studies that have used the same particle tracking module.

Authors' response: Agree. Reviewer#1 has also recommended further improvements to the manuscript in order to provide a more extended and detailed description of the particle transport model. Accordingly, we have rewritten the sub section 3.5 Particle transport model and now reads as follows:

"The application of the transport module of the TESEO particle-tracking model (Abascal et al., 2007, 2017a, b; Chiri et al., 2020) was twofold: (1) simulate the transport and fate of floating litter items once they arrived to the open waters of the SE Bay of Biscay and (2) estimate a windage coefficient by calibrating the model according to the 'low-cost buoys' trajectories. This module allows for simulating passive particles driven by surface currents, wind and turbulent diffusion. Particle trajectories were calculated using the following equation:

$$\frac{d\vec{x}i}{dt} = \vec{u}_a(\vec{x}_1, t) + \vec{u}_d(\vec{x}_1, t)$$
(1)

where  $\vec{u_a}$  and  $\vec{u_d}$  are the advective velocity and diffusive velocity, respectively, for the  $\vec{x_i}$  point and t time. The advective velocity is calculated as the lineal combination of the wind and currents according to:

$$\overrightarrow{u_a} = \overrightarrow{u_c} + C_d \overrightarrow{u_w}$$
(2)

where  $\overline{u_c}$  is the surface current velocity,  $\overline{u_w}$  is the wind velocity at 10m over the sea surface and Cd is the wind drag coefficient. The turbulent diffusive velocity is obtained using Monte Carlo sampling in the range of velocities  $[-\overline{u_{d,}}, \overline{u_d}]$  which are assumed to be proportional to the diffusion coefficients (Hunter et al., 1993; Maier-Reimer and Sündermann, 1982). For each timestep  $\Delta t$ , the velocity fluctuation is defined as:

$$|\overrightarrow{\mathbf{u}_{d}}| = \sqrt{\frac{6\mathrm{D}}{\Delta \mathrm{t}}} \tag{3}$$

where D is the diffusion coefficient, whose value is 1 m2/s in accordance to previously modelling work for floating marine litter (Pereiro et al., 2019; Ruiz et al., 2022). Simulations were forced by HF radar surface current velocity and wind data and interpolated at the particle position for integrating the trajectories. Beaching along the coast was implemented by a simple approach: if the particle reaches the shoreline, it is identified as beached and it is removed from the computational process. TESEO has been calibrated and validated by comparing virtual particle trajectories to observed surface drifter trajectories at regional and local scale (Abascal et al., 2009, 2017a, b; Chiri et al., 2019). Although the TESEO is a 3D numerical model conceived to simulate the transport and degradation of hydrocarbons, it has also been successfully applied to other applications such as the study of transport and accumulation of marine litter in estuaries (Mazarrasa et al., 2019; Núñez et al., 2019) and in open waters (Ruiz et al., 2022)."

**5. Discussion**

The various sections of the discussion seem very disconnected to me. I encourage the authors to streamline the discussion and bring together the multiple aspects of the study. Please try to connect the different aspects of the study (litter distribution, particle tracking and windage) in a better way in the discussion. Regarding the limitations of the model, there are some other problems besides the points raised by the authors. For me, some points remain very unclear. How are the data sets for currents and wind assimilated? What effect does diffusivity have on the pathways of particles in the model or on litter or drifters in the ocean? Does a 4-hour reinitialization of particles suppress tidal effects? All of these questions should be carefully discussed and considered. This is especially important for coastal areas where complex submesoscale processes, fronts, and strong tidal currents become important for particle transport. In addition, Stokes drift is significant for transporting floating objects in the ocean. This should also be discussed in this section.

Authors' response: Thank you for your comments. As the reviewer suggested, we have restructured the manuscript by deleting the sub section 5.6 to achieve a more straightforward and connected discussion. However, we believe that keeping separate sub sections would be more appropriate than bringing all together in order to gain a better understanding of the key aspects of the study. Reviewer#1 has also stated that a more detailed discussion on the limitation of the model would improve the manuscript. Accordingly, we have rewritten this sub section and now reads as follows:

"The interaction between floating litter and the shoreline is highly complex and relies in many processes including waves and tides. Indeed, waves and tides can constrain coastal accumulation since they can resuspend and transport litter back into the ocean (Brennan et al., 2018; Compa et al., 2022). The geomorphology can also affect the retention of litter washing ashore. Sandy beaches tend to be more efficient at trapping and accumulating litter than rocky areas, which favor litter fragmentation (Robbe et al., 2021; Weideman et al., 2020). How these processes contribute to the actual beaching is unknown and they cannot be resolved yet at a suitable resolution (Melvin et al., 2021). In this study, particles were released in open waters and once they reached the shoreline, they were classified as beached. The tidal effect and the wave-induced Stokes drift were not accounted for to avoid introducing more uncertainties. However, further field and laboratory experiments to better understand on how these processes influence floating litter behaviour in the coastline is recommend. It is also important to consider for future research exploring the effect of the type of shoreline on coastal accumulation. In this study, a constant diffusion coefficient of 1 m2/s was considered as a pragmatic choice based on previously modelling work for floating marine litter. However, more field measurements are necessary to accurately assess the influence of the diffusion process on the transport of floating marine litter."

**Specific comments**

I do not want to make remarks about linguistic and spelling mistakes. There are some significant spelling errors such as "week" instead of "weak" or "self-currents," which probably means "shelf-currents". I encourage the authors to carefully revise the manuscript for language and spelling if they decide to resubmit it.

Authors' response: Thank you for your comment. As previously mentioned, we have conducted a deep revision to rephrased long sentences difficult to understand and to amend spelling and grammatical errors pointed out by the reviewers. We hope that this new version is now more suitable for publication.

Figure 5: The authors mention in the caption "trapezoidal integration" I can't find this in the methods chapter. Please explain this in-depth in the methods section as well.

Authors' response: We agree with this comment. We have revised the text consequently to include a short detailed description of the method. The new sentence now reads as follows:

"The area  $\widetilde{D}$  was calculated as a numerical integration over the forecast period via the trapezoidal method following Eq. (5). This method approximates the integration over an interval by breaking the area down into trapezoids with more easily computable areas."

**Please use consistent upper- and lower case in subsection headings.**

Authors' response: Agree. Thank you for your suggestion. We have therefore amended the headings to be consistent with the style and format of the manuscript.

**In line 127, a figure from another publication is cited. This should be avoided.**

Authors' response: Thank you for the suggestion. Reviewer#1 has also stated that citing a figure form other paper is very strange, so we have deleted to provide more understandable information to the reader.

**Lines 115 and 311 are contradictory.**

Authors' response: Thank you for your comment. We have revised both lines and we did not find contradictions between the description of the tidal currents in the study area and the hydrological (geo)morphological characteristics of the rivers. We would appreciate a more detailed explanation from the reviewer to accurately address the comment.

Section 5.6 contains a lot of information about visual observations of litter with camera systems. For me, this is not related to the results of this study. If I understand it correctly, the study was conducted as part of the LIFE-LEMA project. This is also mentioned for the first time in this section and it is confusing to mention it here. Why is the camera system data not included in this study if the project also collected this data? I would recommend including the data or not mentioning it in this section.

Authors' response: Thank you for the suggestion. This specific point was also raised by Reviewer#1. Since this sub section may seem disconnected, we have accordingly removed it to make a clearer and much straightforward discussion.